# In Situ Argon Isotope Analyses of Chondrule-Forming Materials in the Allende Meteorite: A Preliminary Study for $^{40}$Ar/$^{39}$Ar Dating Based on Cosmogenic $^{39}$Ar

**Yuko Takeshima** [1,†]**, Hironobu Hyodo** [1,2]**, Tatsuki Tsujimori** [3]**, Chitaro Gouzu** [4] **and Tetsumaru Itaya** [4,5,6,*]

1 Graduate School of Science, Okayama University of Science, Okayama 700-0005, Japan
2 Institute of Frontier Science and Technology, Okayama University of Science, Okayama 700-0005, Japan
3 Center for Northeast Asian Studies, Tohoku University, Aoba, Sendai 980-8576, Japan
4 Hiruzen Institute for Geology & Chronology, 2-12 Nakashima, Naka-ku, Okayama 703-8252, Japan
5 Japan Geochronology Network, 2-12 Nakashima, Naka-ku, Okayama 703-8252, Japan
6 Institute of GeoHistory, Japan Geochronology Network, 1599 Susai, Akaiwa 701-2503, Japan
* Correspondence: tetsumaru.itaya@gmail.com
† Present address: JAL Brand Communications Co. Ltd., 2-4-11 Higashi-Shinagawa, Shinagawa-ku, Tokyo 140-8643, Japan.

**Abstract:** The argon isotopic compositions of chondrule-forming minerals of the Allende (CV3) meteorite were examined to evaluate the possibility of in situ $^{40}$Ar/$^{39}$Ar dating of planetary surface rocks based on cosmogenic $^{39}$Ar without neutron irradiation in a reactor. The investigated Allende meteorite sample (ME-247H: 50 mm × 45 mm × 5 mm; 28.85 g) contains at least three textural types of chondrules: barred olivine chondrule (BOC), porphyritic olivine chondrule (POC), and unclassified chondrule (UC). Most chondrules contain olivine, low-Ca pyroxene, clinopyroxene, and plagioclase as primary phases, with minor amounts of nepheline and sodalite formed during aqueous alteration of the CV3 parent body of the early solar system. In situ argon isotope analyses on selected chondrule-forming minerals in petrographic sections of two BOCs, two POCs, and one UC using a Nd:YAG pulse laser confirmed a significant amount of cosmogenic $^{39}$Ar that formed by a $^{39}$K (n, p) $^{39}$Ar reaction in an extraterrestrial environment. Laser step-heating analyses of five bulk chondrules irradiated in a reactor revealed a plateau age (3.32 ± 0.06 Ga) from one of the five chondrules. The age spectra of all chondrules show the younger age in the low-temperature fractions, resulting in the integrated ages from 2.7 to 3.2 Ga. These results suggest that the Allende meteorite experienced argon isotopic homogenization at 3.3 Ga and the argon loss in part after the 3.3 Ga. Apparent ages of chondrule-forming minerals that were calculated using the J values of nephelines in one BOC and two POCs do not show any consistent relationship among the three types of chondrules (BOC, POC, and UC). This might be attributed to the fact that the isotopic heterogeneity among minerals took place during the heterogeneous argon loss stage after the 3.3 Ga event.

**Keywords:** chondrules; Allende meteorite; in situ argon isotope analyses; $^{40}$Ar/$^{39}$Ar dating

## 1. Introduction

In the cosmic space outside the Earth, high-energy cosmic ray irradiation can cause a $^{39}$K (n, p) $^{39}$Ar reaction in K-bearing materials [1]. If the cosmogenic $^{39}$Ar generated by the process can be detected, it would enable the dating of extraterrestrial samples during landing planetary missions. For example, the application could be used for lunar surface rocks by direct in situ $^{40}$Ar/$^{39}$Ar measurements with equipment mounted on the lunar explorer based on the information on the K–Ar age and $^{40}$Ar/$^{39}$Ar ratio of the lunar samples collected by the Apollo mission. This perspective may contribute to reducing the uncertainty of the inferred age of the lunar crater chronology from one billion years to 0.1 billion years of errors because of the in situ $^{40}$Ar/$^{39}$Ar dating of planetary surface rocks

based on cosmogenic $^{39}$Ar without neutron irradiation in a reactor. With the same objective, in situ K–Ar isochron dating was tested for terrestrial silicate minerals using a spot-by-spot laser ablation technique by Cho et al. (2015, 2016) [2,3].

As a pilot study for the evaluation of dating planetary surface materials based on cosmogenic $^{39}$Ar, Takeshima (2001) [4] carried out argon isotope analysis of bulk chondrules (~1.7–3.5 × 10$^{-7}$ g) of the Allende meteorite and succeeded in detecting a considerable amount of cosmogenic $^{39}$Ar (~1.3–3.0 × 10$^{-12}$ ccSTP/g within 7%–20% error). However, because the investigated chondrules contain secondary Cl-bearing minerals that were formed by aqueous alteration on the meteorite parent body (e.g., [5,6]), the radiogenic $^{40}$Ar would likely be underestimated due to the cosmogenic $^{36}$Ar produced by neutron capture on $^{35}$Cl, yielding younger $^{40}$Ar/$^{39}$Ar ages. Moreover, $^{35}$Cl and $^{37}$Cl in secondary minerals can form some hydrogen chloride molecules in a mass spectrometer; note that the hydrogen chlorides are detected as isobars of $^{36}$Ar and $^{38}$Ar, which are indistinguishable in a mass spectrometer with a resolution of less than 3000. To evaluate the distribution of Cl-bearing minerals against the K-bearing phases in chondrules of the Allende meteorite, Takeshima et al. (2003) [7] performed mineralogical observation and confirmed significant chemical heterogeneities of chlorine and potassium in the chondrules that potentially obstruct accurate argon isotope analyses. Naturally, their reconnaissance led to the conclusion that the petrographical analyses with elemental mappings for chondrules were crucial for precise in situ argon isotope analyses.

In this paper, we report the results of in situ argon isotope analyses of the minerals that are present in chondrules (from now on, chondrule-forming minerals) of the Allende meteorite, with a special focus on the spatial distribution of K-bearing phases such as nepheline and Cl-bearing minerals such as sodalite, and show the results of laser step-heating $^{40}$Ar/$^{39}$Ar analyses of the bulk chondrules irradiated in a reactor. To calculate the ages of chondrule-forming minerals using the J values of nephelines as an inferred standard, we describe the $^{40}$Ar/$^{39}$Ar age determination method using cosmogenic $^{39}$Ar. We also mention the $^{39}$Ar argon recoil problem during high-energy cosmic ray irradiation. This approach can be meaningful not only in chondrule chronology, but also in challenging the in situ $^{40}$Ar/$^{39}$Ar dating of planetary surface rocks based on cosmogenic $^{39}$Ar without neutron irradiation in a reactor.

## 2. Sample Description and EMP Analytical Procedure

We examined a small slab (50 mm × 45 mm × 5 mm, 28.85 g) of the Allende CV3 meteorite sample (ME-427H; purchased from Mineralogical Research Co., San Jose, CA, USA). The slab contains abundant chondrules and some Ca-Al-rich inclusions (CAIs) (Figure 1) and may also contain the amoeboid olivine aggregates (AOAs) studied by Grossman and Steele (1976) [8]. There are various fields of research for this meteorite. Chronologically, it is well known that CAIs are the most primitive materials, with Pb–Pb ages ranging between 4565 ± 1 and 4568 ± 3 Ma [9]. These ages were confirmed by a Pb–Pb isochron age of 4567.72 ± 0.93 Ma defined by six CAIs [10]. The $^{26}$Al–$^{26}$Mg studies of chondrules have revealed the formation of chondrules that took place ~2–3 Myr after the formation of CAIs [11]. Connelly et al. (2012) [12] provided a Pb–Pb isochron age of 4567.32 ± 0.42 Ma using the isotopic data of six chondrules. Doyle et al. (2015) [13] conducted $^{53}$Mn–$^{53}$Cr dating of aqueously formed fayalite in the Asuka 881317 CV3 chondrite similar to the Allende CV3 chondrite and concluded that the fayalite formed 4.2 + 0.8/−0.7 Myr after CAI formation. This result constrains the formation during the aqueous alteration in the CV3 parent body of the early solar system. Miura et al. (2014) [14] reported $^{3}$He cosmic ray exposure ages of 5.17 ± 0.38 Myr and 5.15 ± 0.25 Myr for the averages of the chondrules and whole rocks, respectively.

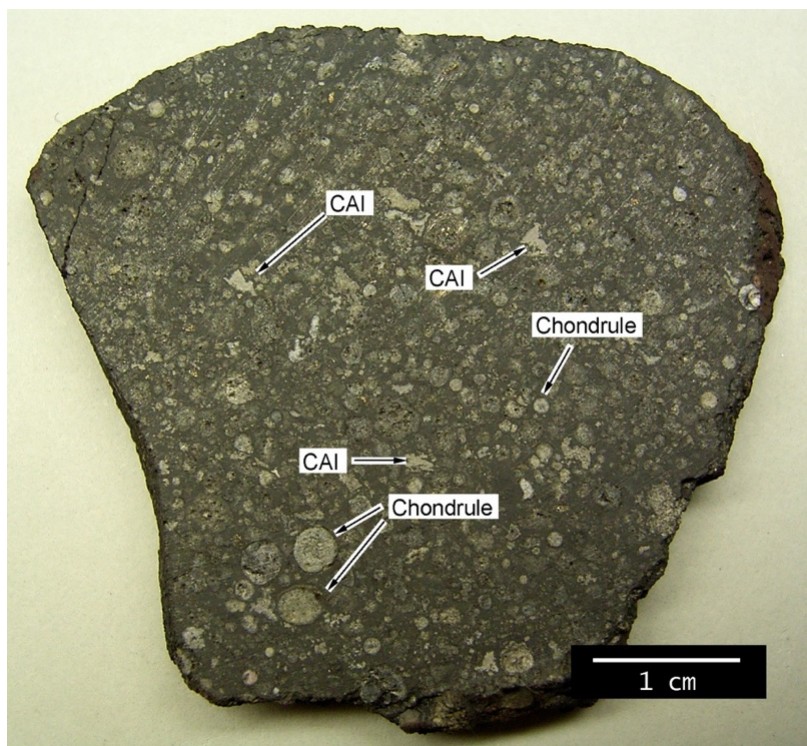

**Figure 1.** Photograph of the investigated Allende meteorite slab (ME-427).

In this study, we examined thirty-seven small objects (chondrules and CAIs) in a polished section (0.1 mm thick) using a JEOL JXA-8900R electron microprobe (EMP) analyzer at Okayama University of Science; back-scattered electron (BSE) imaging and X-ray elemental mappings were performed with 15 kV accelerating voltage and 240 nA beam current. The BSE images and X-ray maps of the thirty-seven objects are shown in Supplementary Figures S1 and S2, respectively. The major element analyses of silicate minerals in chondrules were performed with 15 kV accelerating voltage, 12 nA beam current, and 3 μm beam size; natural and synthetic silicates and oxides were used for calibration; the ZAF method (oxide basis) was employed for matrix corrections. The major element compositions are summarized in Supplementary Table S1.

### 3. Petrography and Mineral Chemistry of Chondrules, Olivine Fragments, and CAIs

Based on the EMP textural observation (Supplementary Figures S1 and S2), the thirty-seven investigated objects were grouped into the barred olivine chondrule (BOC: eleven objects), the porphyritic olivine chondrule (POC: nineteen objects), the unclassified chondrule (UC: three objects), the olivine fragment (OF: two objects), and the Ca-Al-rich inclusion (CAI: two objects). The petrographic descriptions of each object with chemical compositions (Supplementary Table S1) are referred to in the Supplementary Materials (SM). The mineral phases observed in each object are listed in Table 1. Most objects consist mainly of olivine, low-Ca pyroxene, clinopyroxene, and plagioclase, with minor amounts of secondary nepheline and sodalite.

**Table 1.** Mineral phases observed in each object. Mineral abbreviations: olivine—Ol, low-Ca pyroxene—OPX, clinopyroxene—CPX, plagioclase—Pl, spinel—Sp, gehlenite—Geh, nepheline—Ne, sodalite—Sod.

| Sample No. | Ol | CPX | OPX | Pl | Sp | Geh | Ne | Sod |
|---|---|---|---|---|---|---|---|---|
| Barred olivine chondrule | | | | | | | | |
| BOC-01 | ○ | ○ | | ○ | | | ○ | ○ |
| BOC-02 | ○ | | | | | | ○ | ○ |
| BOC-03 | ○ | | ○ | | | | ○ | ? |
| BOC-04 | ○ | | | | ○ | | ○ | ○ |
| BOC-05 | ○ | ○ | ○ | | | | ○ | ○ |
| BOC-06 | ○ | ○ | | ○ | | | ○ | ○ |
| BOC-07 | ○ | ○ | ○ | | | | ○ | ○ |
| BOC-08 | ○ | ○ | | ○ | | | ○ | ○ |
| BOC-09 | ○ | | | | | | ○ | ○ |
| BOC-10 | ○ | ○ | ○ | ○ | | | ○ | ○ |
| BOC-11 | ○ | ○ | | | | | ○ | ? |
| Porphyritic olivine chondrule | | | | | | | | |
| POC-01 | ○ | ○ | | ○ | | | ○ | ○ |
| POC-02 | ○ | ○ | ○ | ○ | | | ○ | ○ |
| POC-03 | ○ | ○ | ○ | ○ | | | ○ | ○ |
| POC-04 | ○ | ○ | | ○ | | | ○ | ○ |
| POC-05 | ○ | ○ | ○ | ○ | | | ○ | ○ |
| POC-06 | ○ | ○ | ○ | | | | ○ | ○ |
| POC-07 | ○ | ○ | ○ | ○ | | | ○ | ○ |
| POC-08 | ○ | ○ | ○ | | | | ○ | ○ |
| POC-09 | ○ | ○ | ○ | | | | ○ | ○ |
| POC-10 | ○ | ○ | ○ | | | | ○ | ○ |
| POC-11 | ○ | ○ | ○ | | | | ○ | ○ |
| POC-12 | ○ | ○ | ○ | | | | ○ | ○ |
| POC-13 | ○ | ○ | ○ | ○ | | | ○ | ○ |
| POC-14 | ○ | | | ○ | | | ○ | ○ |
| POC-15 | ○ | ○ | ○ | ○ | | | ○ | ○ |
| POC-16 | ○ | ○ | ○ | | | | ○ | ○ |
| POC-17 | ○ | | ○ | ○ | | | ○ | ○ |
| POC-18 | ○ | ○ | | | | | ○ | ○ |
| POC-19 | ○ | ○ | | ○ | | | ○ | ○ |
| Unclassified chondrule | | | | | | | | |
| UC-01 | ○ | ○ | ○ | ○ | | | ○ | ○ |
| UC-02 | ○ | ○ | | ○ | | | ○ | ○ |
| UC-03 | ○ | ○ | | ○ | | | ○ | ○ |
| Olivine fragment | | | | | | | | |
| OF-01 | ○ | | | | ? | | ○ | ? |
| OF-02 | ○ | | | | ○ | | ○ | ? |
| CAI | | | | | | | | |
| CAI-01 | | ○ | | | | | ○ | ○ |
| CAI-02 | | ○ | | ○ | ○ | ○ | ? | ? |

Descriptions of symbols: Circle indicates presence. The mark "?" indicates that there may be. No mark indicates no presence.

Figure 2 shows the textures of the representative objects, the barred olivine chondrule (BOC-01), porphyritic olivine chondrule (POC-06), unclassified chondrule (UC-02), and olivine fragment (OF-02).

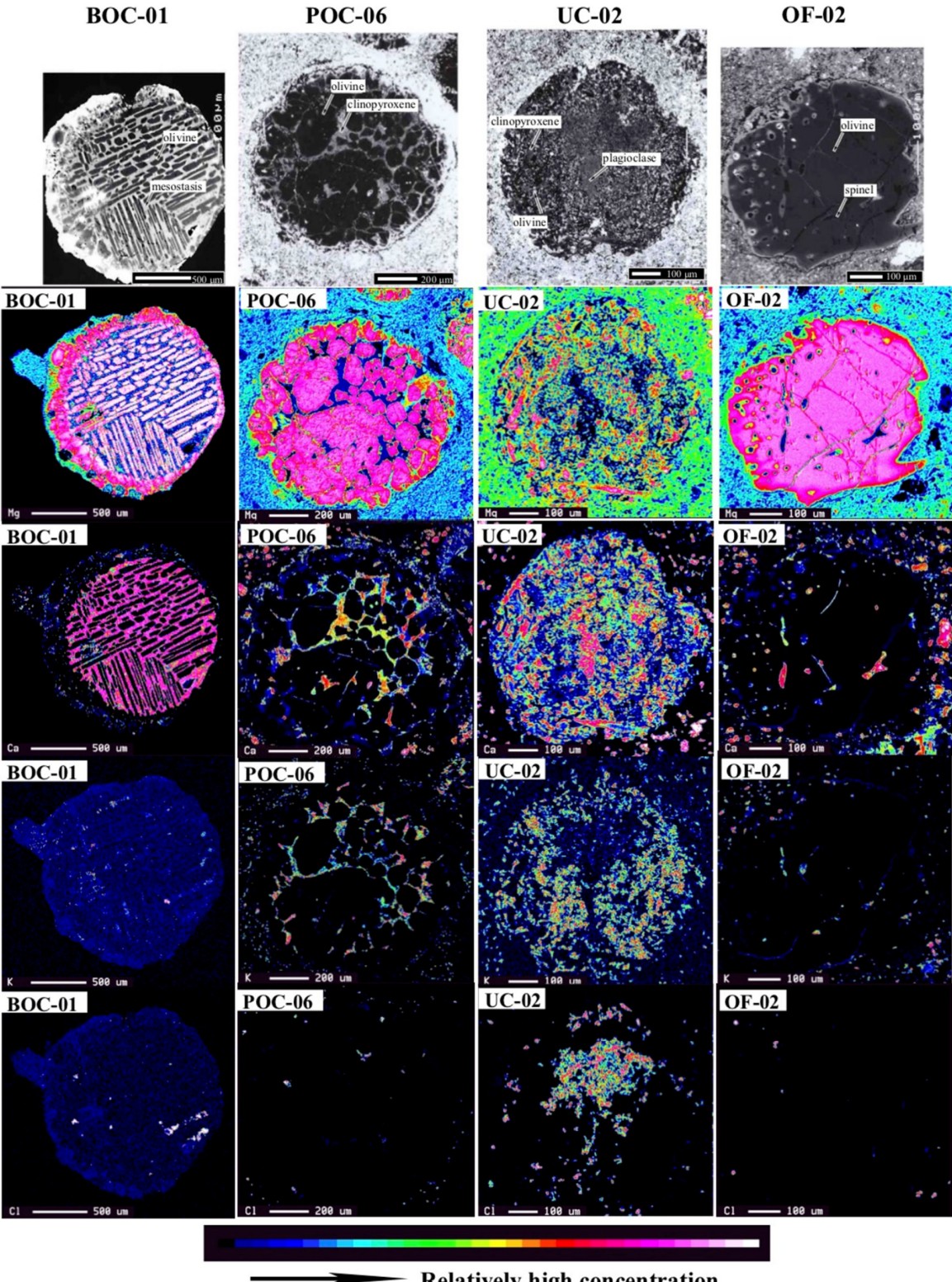

**Figure 2.** BSE images and Mg–Ca–K–Cl X-ray maps of the representative objects, barred olivine chondrule (BOC-01), porphyritic olivine chondrule (POC-06), unclassified chondrule (UC-02), and olivine fragment (OF-02).

BOC-01 consists mainly of barred olivine with plagioclase, with minor amounts of Al-rich clinopyroxene and secondary nepheline and sodalite. Barred olivine is forsterite-rich, with a Mg/(Mg + Fe) atomic ratio [= $X_{Mg}$] ranging from 0.98 to 0.99. The $X_{Mg}$ values tend to decrease from the core part to the rim part of the chondrule (Figure 2). The rim part has $X_{Mg}$ from 0.93 to 0.98 (see Supplementary Table S1). Plagioclase is Ca-rich with a 91–92 mol% anorthite component. Al-rich clinopyroxene contains 3.5–8.1 wt% $Al_2O_3$, 1.5–2.1 wt% $TiO_2$, 0.39–0.55 wt% $Cr_2O_3$, and 0.08–0.14 wt% $Na_2O$ with $X_{Mg}$ = 0.98 to 0.99. Nepheline-replacing mesostatic plagioclase contains up to 1.8 wt% $K_2O$. Sodalite is closely associated with nepheline (Figure 3).

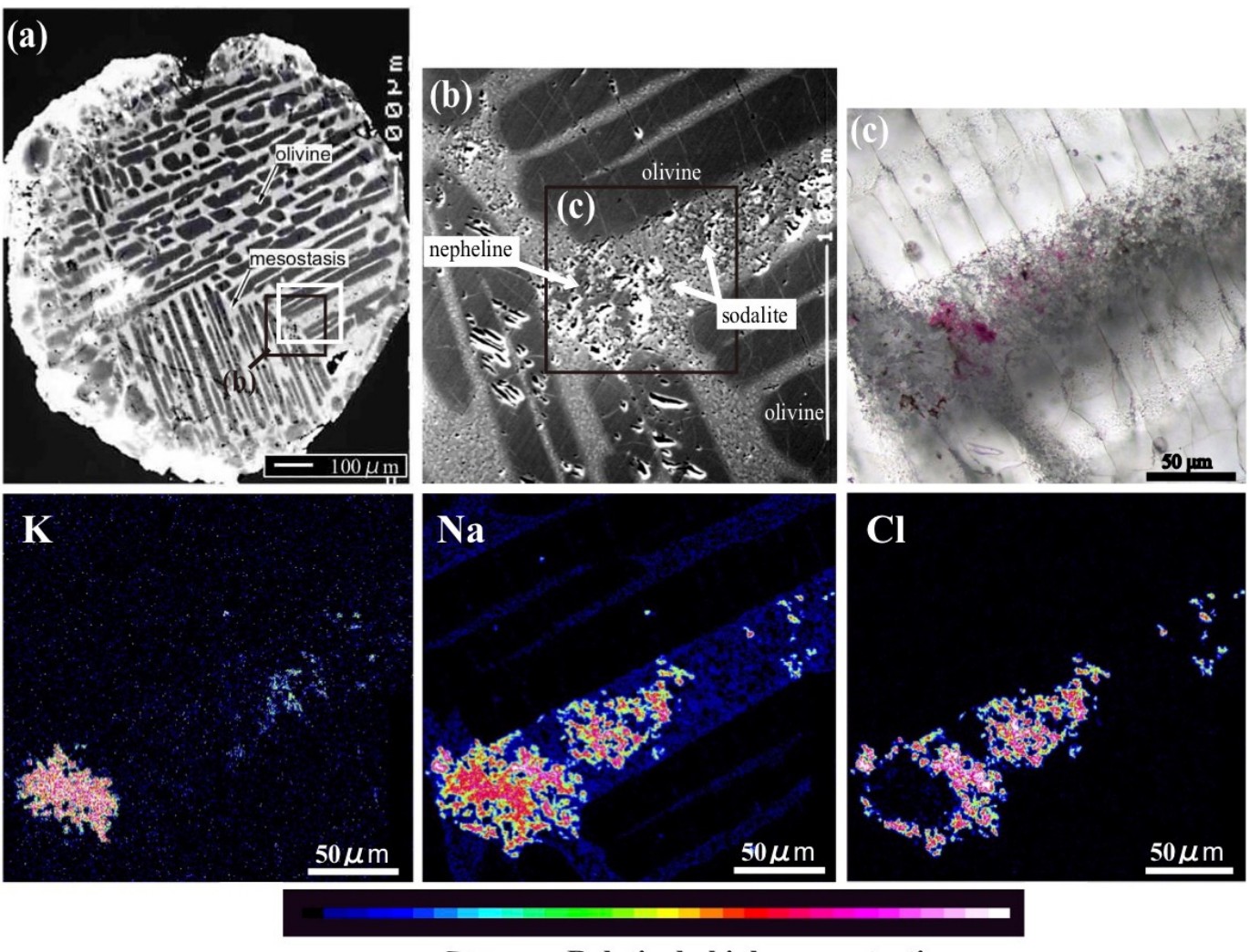

**Figure 3.** (**a**) BSE image of barred olivine chondrule (BOC-01). (**b**) Enlarged BSE image of the black boxed area in (**a**), showing the occurrence of nepheline and sodalite in mesostasis. White line is a scale of 100 microns. (**c**) Optical photomicrograph (plane-polarized view) of a boxed area of (**b**), showing pinkish sodalite. K–Na–Cl X-ray maps of the white boxed area in (**a**) are also shown.

POC-06 contains mainly olivine, low-Ca pyroxene, and Al-rich clinopyroxene, with secondary nepheline and minor amounts of sodalite. Olivine is mainly forsterite ($X_{Mg}$ = 0.91–0.99 (see Supplementary Table S1)). Olivine of the rim part of the chondrule (Figure 2) gives 0.64 of $X_{Mg}$. Low-Ca pyroxene contains 1.1 wt% $Al_2O_3$, 0.5 wt% CaO, and 0.5 wt% $Cr_2O_3$ with $X_{Mg}$ = 0.99. Al-rich clinopyroxene contains 5.5–14.8 wt% $Al_2O_3$, 0.5–1.6 wt% $TiO_2$, 1.02–1.36 wt% $Cr_2O_3$, and 0.03–1.29 wt% $Na_2O$; $X_{Mg}$ ranges from 0.96 to 0.99. Nepheline contains 1.6–2.0 wt% $K_2O$.

UC-02 consists mainly of olivine, clinopyroxene, and plagioclase, with significant amounts of nepheline and sodalite (Figure 2). Olivine is Fe-rich ($X_{Mg} = 0.69$) (see Supplementary Table S1). Clinopyroxene contains 2.4–4.0 wt% $Al_2O_3$, 4.8–18.5 wt% CaO, 0.4–2.0 wt% $TiO_2$, 0.4–0.7 wt% $Cr_2O_3$, and <0.7 wt% $Na_2O$; $X_{Mg}$ varies from 0.93 to 0.99. Plagioclase contains a 97 mol% anorthite component. Nepheline contains 1.4–1.9 wt% $K_2O$.

OF-02 consists mainly of olivine and Mg-Al spinel. Olivine shows distinct chemical zoning, and the forsterite-rich cores ($X_{Mg} = 0.97$–1.00) are rimmed by Fe-rich olivine ($X_{Mg} = 0.59$–0.75). Mg-Al spinel contains 0.63 wt% $Cr_2O_3$, and its $X_{Mg}$ is 0.94 (see Supplementary Table S1).

Secondary nepheline and sodalite in BOC-01 are closely associated with each other (Figure 3). This association is commonly observed in other objects, as determined from K–Na–Cl X-ray maps of the objects (see Supplementary Figure S2). The modal abundance of nepheline and sodalite is variable among the objects. However, their abundances are generally higher in the former than in the latter; see the X-ray maps of POC-03, POC-012, POC-13, POC-15, POC-16, UC-01, and UC-02 summarized in Supplementary Figure S2. Olivine fragments also contain small amounts of nepheline and sodalite.

It has been known that nepheline and sodalite in the Allende meteorite were formed by aqueous alteration in the parent body [5,6,15–19] (see Supplementary Materials).

## 4. $^{40}$Ar/$^{39}$Ar Method

The $^{40}$Ar/$^{39}$Ar method uses the same radiation decay system as the K-Ar method. However, it has various advantages that the K-Ar method does not have due to its devised measurement method. The first attempt at $^{40}$Ar/$^{39}$Ar dating was made by Sigurgeirsson [20] at the University of Iceland in 1962. This method began to attract attention after it was applied to meteorites and shown to be effective in elucidating their thermal history [21,22]. Turner contributed to the establishment of the foundation of the $^{40}$Ar/$^{39}$Ar method [23,24]. This method provided highly reliable age results for the dating of lunar rocks brought back to Earth by the Apollo program, and its effectiveness was demonstrated [cf. [25]].

For the $^{40}$Ar/$^{39}$Ar dating, the sample is irradiated in a nuclear reactor with fast neutrons (E > 1MeV), inducing the reaction $^{39}$K (n, p) $^{39}$Ar. Since $^{39}$Ar has a half-life of 269 years, it hardly decreases during the period from neutron irradiation to Ar isotope measurement. On the other hand, Ar isotopes are also generated by nuclear reactions other than the reaction $^{39}$K (n, p) $^{39}$Ar. For the correction, the Ca and K compounds are also irradiated with neutrons at the same time, and the correction coefficient for the Ar isotope formed from the Ca and K irradiated is determined. The age (t) is obtained using the $^{40}$Ar/$^{39}$Ar ratio corrected for the Ar isotopes formed from the Ca and K irradiated, and the J value is obtained from the standard sample which has the K-Ar age ($t_s$) using Equation (1).

$$t = \ln(1 + J^{40}Ar/^{39}Ar)/\lambda \tag{1}$$

The J value is obtained from the following equation.

$$J = (\exp(\lambda t_s) - 1)/(^{40}Ar/^{39}Ar)_s \tag{2}$$

$(^{40}Ar/^{39}Ar)_s$ in this equation is the $^{40}$Ar/$^{39}$Ar ratio of a standard sample which has the K-Ar age ($t_s$).

Therefore, the standard sample is also irradiated with fast neutrons together with the samples and the Ca and K compounds [26].

As discussed later, the same Equations (1) and (2) are used in the in situ $^{40}$Ar/$^{39}$Ar dating of planetary surface rocks based on cosmogenic $^{39}$Ar without neutron irradiation in a reactor.

## 5. Argon Isotope Analyses

### 5.1. Mass Spectrometer

The mass spectrometer and the associated analytical systems were originally designed and constructed to analyze noble gas isotopes in meteorites by K. Ogata and K. Nagao at Okayama University of Science in the late 1970s. The mass spectrometer is composed of similar elements to the mass spectrometer of Takaoka (1976) [27]: a sample holder, an extraction oven, purification lines, standard gas lines, a mass spectrometer, and an ultra-high vacuum pumping system. All of them, except for the glass sample holder, were made of stainless steel and connected with the ICF flange using Cu gaskets or ultra-high vacuum metal valves (see [28] Figure 4). The inner surface of the stainless steel of the mass spectrometer, extraction–purification lines, and standard gas lines were treated with electropolishing. All stainless steel parts are bakeable using glass-fiber-insulated nickel chromium heaters. The extraction–purification lines and standard gas lines are pumped out by turbo molecular pumps to $10^{-9}$ Torr, and the mass spectrometer is pumped out by a sputter ion pump to $10^{-10}$ Torr. A series of analyses by the mass spectrometer, such as taking a set of mass spectra and the calculation of their ratios, and isotope content is done with a computer-controlled system. This low-blank, high-sensitivity, and high-resolution mass spectrometer with a computer-controlled system contributed to the development of noble gas geochemistry and cosmochemistry [cf. [29–33]].

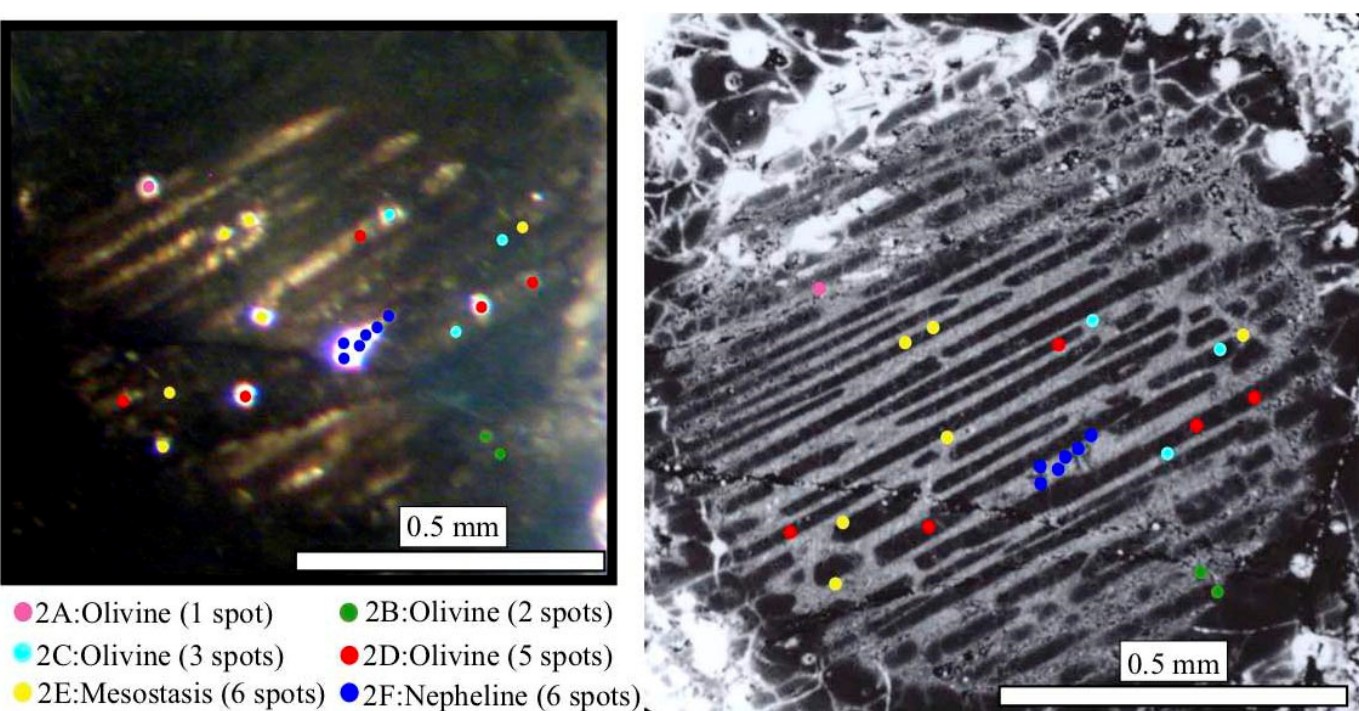

**Figure 4.** Optical view and its BSE image of BOC-06 showing laser shot locations with different colors. The number of laser spots is one shot for the 2A olivine phase (pink), two shots for the 2B olivine phase (green), three shots for the 2C olivine phase (sky blue), five shots for the 2D olivine phase (red), and six shots for the 2E mesostatic phase (yellow) and 2F nepheline phase (blue).

The sample extraction–purification line and the detector of argon analyses were improved for the $^{40}Ar/^{39}Ar$ dating of submillimeter-sized single mineral grain [34,35]. The mass spectra of blank analyses confirmed the resolution of the mass spectrometer was ca. 800 (see [34] Figure 2). This resolution makes it to possible to completely separate the interfering hydrocarbon ions from the argon isotopes of $^{37}Ar$, $^{38}Ar$, $^{39}Ar$, and $^{40}Ar$. They also revealed that the blanks of the purification line are $1 \times 10^{-14}$, $1 \times 10^{-13}$, and $3 \times 10^{-12}$ ccSTP for $^{36}Ar$, $^{39}Ar$, and $^{40}Ar$, respectively. Two types of laser-heating systems were used in the gas extraction: a Nd:YAG pulse-laser for the in situ spot analyses of

single minerals and a 5 W continuous argon-ion laser for step-heating analyses of the individual chondrules. This new mass spectrometric system for the $^{40}Ar/^{39}Ar$ dating of submillimeter-sized single mineral grain and in situ dating has been applied to the various geological samples [cf. [36–43]].

*5.2. In Situ Argon Isotope Analyses of Minerals in the Chondrules*

After EMP petrographical analyses, five portions containing objects BOC-06, BOC-07, POC-13, POC-06, and UC-02 were removed from a 0.1 mm thick polished section for Nd:YAG pulse-laser analyses. The spot size of the pulse laser was ~20 microns; the extracted gas was purified with a SAES Zr–Al getter (St 101) kept at 400 °C for 5 min. Argon isotopes were measured using the mass spectrometer mentioned above. Mass discrimination was checked with atmospheric argon each day. The blanks of argon isotopes were checked before each sample analysis.

Olivine in BOC-06 was analyzed using the sample gas of the different numbers of laser spots (1, 2, 3, and 5). Nepheline and mesostasis phases in the same sample were also analyzed using the sample gas of six laser spots. Minerals of other samples were analyzed using the sample gas of ten spots. The results are shown in Table 2.

Figures 4–8 show the laser shot locations of objects BOC-06, BOC-07, POC-13, POC-06, and UC-02, respectively. Table 2 clearly shows significant amounts of cosmogenic $^{39}Ar$ formed by a $^{39}K$ (n, p) $^{39}Ar$ reaction. It should be noted that olivine, which cannot accommodate K in its crystal structure, has a detectable amount of cosmogenic $^{39}Ar$. This might be due to K as a chemical impurity in olivine or contamination by the adjacent minerals. In fact, some olivine data show a small amount of $K_2O$ (0.08 wt% in maximum), as shown in Supplementary Table S1. On the other hand, olivine has a significant amount of CaO, up to 0.8wt% (see Supplementary Table S1). This means the cosmogenic $^{39}Ar$ may be in part formed by the reaction of $^{42}Ca(n, \alpha)^{39}Ar$. Olivine + pyroxene data with a relatively high $^{39}Ar$ in BOC-07 might be attributed to the same reason. K-bearing nepheline and sodalite have $^{40}Ar$ amounts that are significantly higher than that of olivine. In POC-13, sodalite with lower K content has a cosmogenic $^{39}Ar$ value ($48.0 \times 10^{-11}$ ccSTP) higher than that ($16.5 \times 10^{-11}$ ccSTP) of K-rich nepheline. This enrichment is most likely the result of contamination of high-K nepheline during sodalite analysis (see Figure 3). The $^{38}Ar/^{36}Ar$ ratios of minerals in the investigated chondrules range from 0.20 to 0.81, except for olivine, which has extremely high ratios up to 2.0 (Table 2). These isotope variations are similar to those (0.19 to 0.81) of ten chondrules of Allende in a previous study [14]. They interpreted that low-temperature fractions of three chondrules with $^{38}Ar/^{36}Ar$ ratios lower than 0.19 are due to the addition of $^{36}Ar$ produced by neutron capture on $^{35}Cl$.

Figure 9 shows a plot of $^{38}Ar$ versus $^{36}Ar$ of the analyzed chondrule-forming minerals. The solid and dashed lines in the figure are $^{38}Ar/^{36}Ar$ = 1/3 and $^{38}Ar/^{36}Ar$ = 1/5, respectively. The $^{38}Ar/^{36}Ar$ ratios of sodalite are plotted on the solid line. Since the extracted gas was purified, this would not be due to the formation of hydrogen chlorides during the isotope analyses. The cosmogenic reactions of $^{35}Cl$ (n, γ) $^{36}Cl$, β⁻ decay to $^{36}Ar$, and $^{37}Cl$ (n, γ) $^{38}Cl$, β⁻ decay to $^{38}Ar$, giving the ratio much smaller than the natural ratio of $^{37}Cl/^{35}Cl$ = 1/3 because of the difference of the thermal neutron cross-sections for $^{37}Cl$ and $^{35}Cl$ [44]. The sodalite has 0.3–2.8 wt% of CaO (see Supplementary Table S1), suggesting that the cosmogenic $^{36}Ar$ formed by the reaction $^{40}Ca$ (n, nα) $^{36}Ar$. This makes the ratio much lower. On the other hand, the cosmogenic $^{38}Ar$ can form by the reactions $^{42}Ca$ (n, nα) $^{38}Ar$ and $^{41}K$ (nα, β⁻) $^{38}Ar$. BOC−06 olivine of analytical number 1210A2B gives a high $^{38}Ar/^{36}Ar$ ratio of 2.0, which is an order of magnitude higher than other olivine analyses. It might be a localized record of spallogenic argon [cf. [44]].

**Table 2.** In situ argon isotope data of minerals in the investigated chondrules of the Allende meteorite. The number in parentheses after the sample name indicates the total number of laser spots. The abbreviations of olivine, pyroxene, nepheline, sodalite, and mesostasis are Ol, Py, Neph, Soda, and Meso, respectively. See the later Section 6 for the meaning of $(^{40}Ar/^{39}Ar)_{analyzed}$, $(^{40}Ar/^{39}Ar)_{corrected}$, J values, and ages (text colors in red, blue, and purple). All error values represent 1σ.

| Analitical | Sample | $^{36}Ar$ | $^{38}Ar$ | $^{39}Ar$ | $^{40}Ar$ | $^{40}Ar/^{36}Ar$ | $^{38}Ar/^{36}Ar$ | $(^{40}Ar/^{39}Ar)_a$ | $(^{40}Ar/^{39}Ar)_c$ | J value | Age | Age | Age |
|---|---|---|---|---|---|---|---|---|---|---|---|---|---|
| Numbers | Numbers | $10^{-14}$(ccSTP) | $10^{-14}$(ccSTP) | $10^{-14}$(ccSTP) | $10^{-11}$(ccSTP) | $10^{+2}$ | | $10^{+2}$ | $10^{+2}$ | $10^{+3}$ | (Ga) | (Ga) | (Ga) |
| 1210A2A | BOC-06Ol(1) | 4.25 ± 0.43 | 1.22 ± 0.37 | 3.15 ± 1.21 | 2.86 ± 0.12 | 6.71 | 0.287 | 9.05 | 9.91 | | 2.59 | 4.28 | 1.06 |
| 1210A2B | BOC-06Ol(2) | 1.71 ± 0.46 | 3.49 ± 0.52 | 7.03 ± 1.80 | 2.55 ± 0.09 | 14.9 | 2.04 | 3.63 | 3.97 | | 1.49 | 2.87 | 0.5 |
| 1210A2C | BOC-06Ol(3) | 5.55 ± 0.44 | 2.35 ± 0.32 | | 8.00 ± 0.17 | 14.4 | 0.424 | | | | | | |
| 1210A2D | BOC-06Ol(5) | 6.84 ± 0.40 | 3.63 ± 0.41 | 2.28 ± 4.37 | 13.5 ± 0.40 | 19.7 | 0.531 | 59.2 | 64.8 | | 5.57 | 7.53 | 3.30 |
| 1210A2E | BOC-06Meso | 10.8 ± 0.47 | 4.19 ± 0.43 | 11.0 ± 2.04 | 13.1 ± 0.44 | 12.1 | 0.386 | 11.9 | 13.1 | | 2.98 | 4.74 | 1.30 |
| 1210A2F | BOC-06Neph | 5.59 ± 0.56 | 2.58 ± 0.50 | 0.925 ± 2.81 | 12.1 ± 0.56 | 21.7 | 0.463 | 131 | 143 | 80.6 | 6.96 | 8.94 | 4.56 |
| 1210A2G | BOC-07Ol+Py | 28.4 ± 0.63 | 8.85 ± 0.43 | 30.0 ± 4.19 | 27.7 ± 0.89 | 9.75 | 0.311 | 9.23 | 10.1 | | 2.62 | 4.32 | 1.07 |
| 1210A2H | POC-13Soda | 47.6 ± 0.91 | 15.1 ± 0.85 | 48.0 ± 7.52 | 29.8 ± 0.55 | 6.26 | 0.317 | 6.21 | 6.8 | | 2.10 | 3.68 | 0.79 |
| 1210A2I | POC-13Neph | 27.0 ± 0.79 | 13.2 ± 0.76 | 16.5 ± 5.18 | 53.8 ± 2.27 | 19.9 | 0.487 | 32.7 | 35.7 | 3.23 | 4.56 | 6.47 | 2.45 |
| 1210A5A | POC-06Ol | 4.94 ± 0.50 | 4.02 ± 0.54 | 20.2 ± 2.27 | 3.41 ± 0.15 | 6.91 | 0.814 | 1.69 | 1.85 | | 0.85 | 1.87 | 0.25 |
| 1210A5B | POC-06Neph | 16.1 ± 0.71 | 9.46 ± 0.76 | 33.4 ± 7.02 | 35.8 ± 1.70 | 22.2 | 0.587 | 10.7 | 11.7 | 9.85 | 2.83 | 4.56 | 1.20 |
| 1210A5C | UC-02Soda | 57.1 ± 0.86 | 19.7 ± 1.06 | 55.5 ± 6.83 | 92.4 ± 5.62 | 16.2 | 0.345 | 16.7 | 18.2 | | 3.48 | 5.31 | 1.63 |

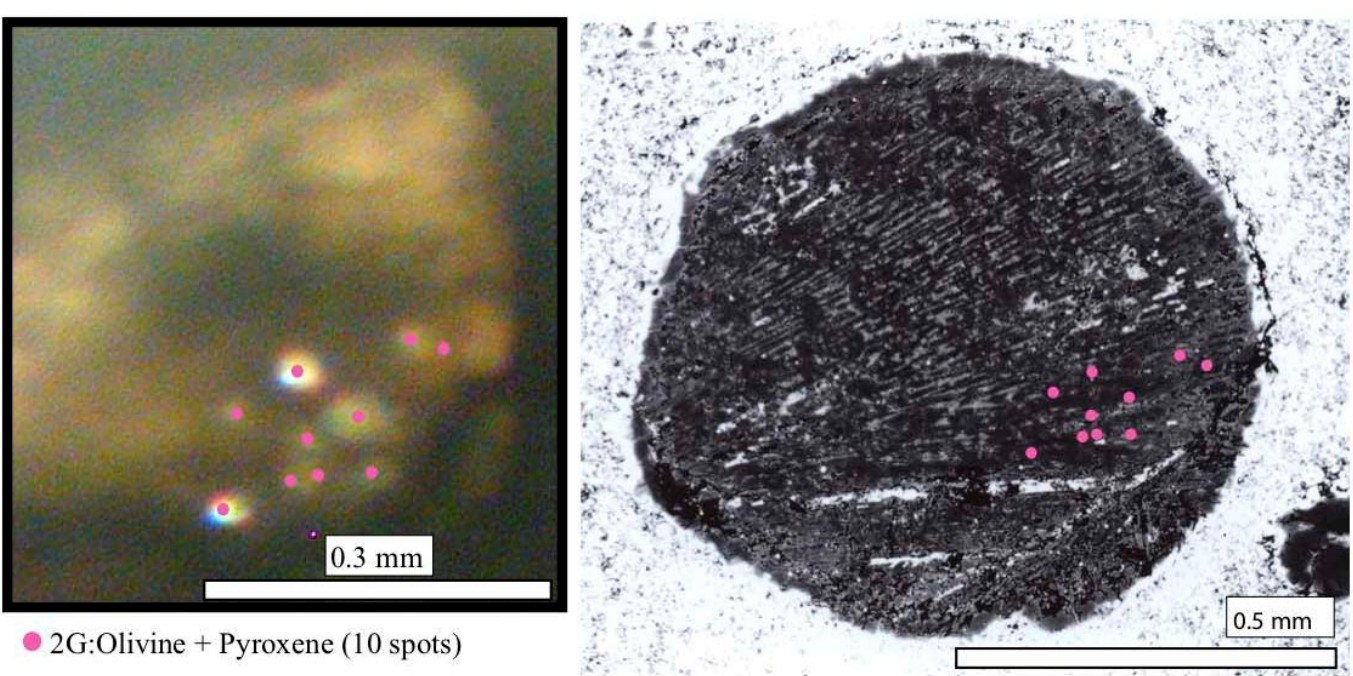

**Figure 5.** Optical view and BSE image of BOC-07 showing laser shot locations. The number of laser spots is ten shots for the 2G olivine + pyroxene phase (pink).

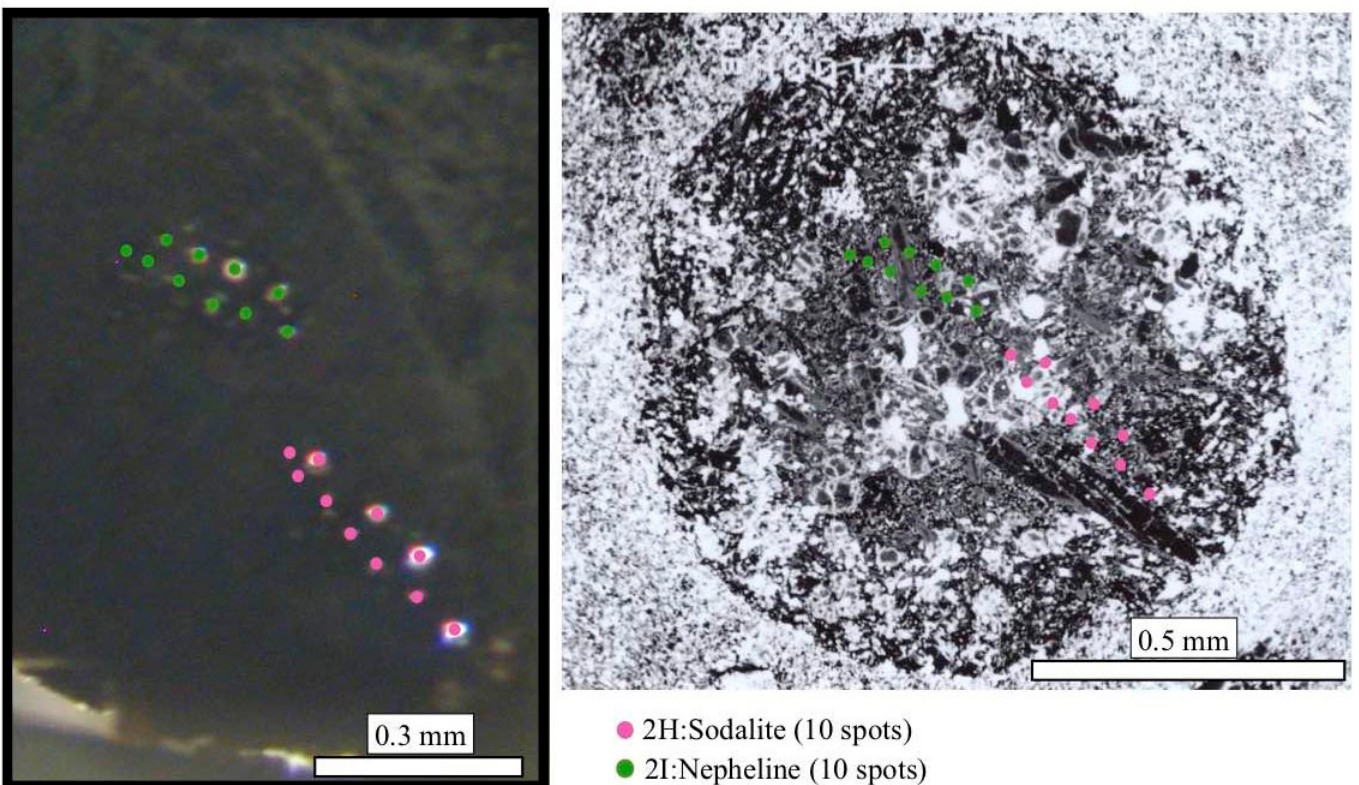

**Figure 6.** Optical view and BSE image of POC-13 showing laser shot locations. The number of laser spots is ten shots for the 2H sodalite phase (pink) and for the 2I nepheline phase (green).

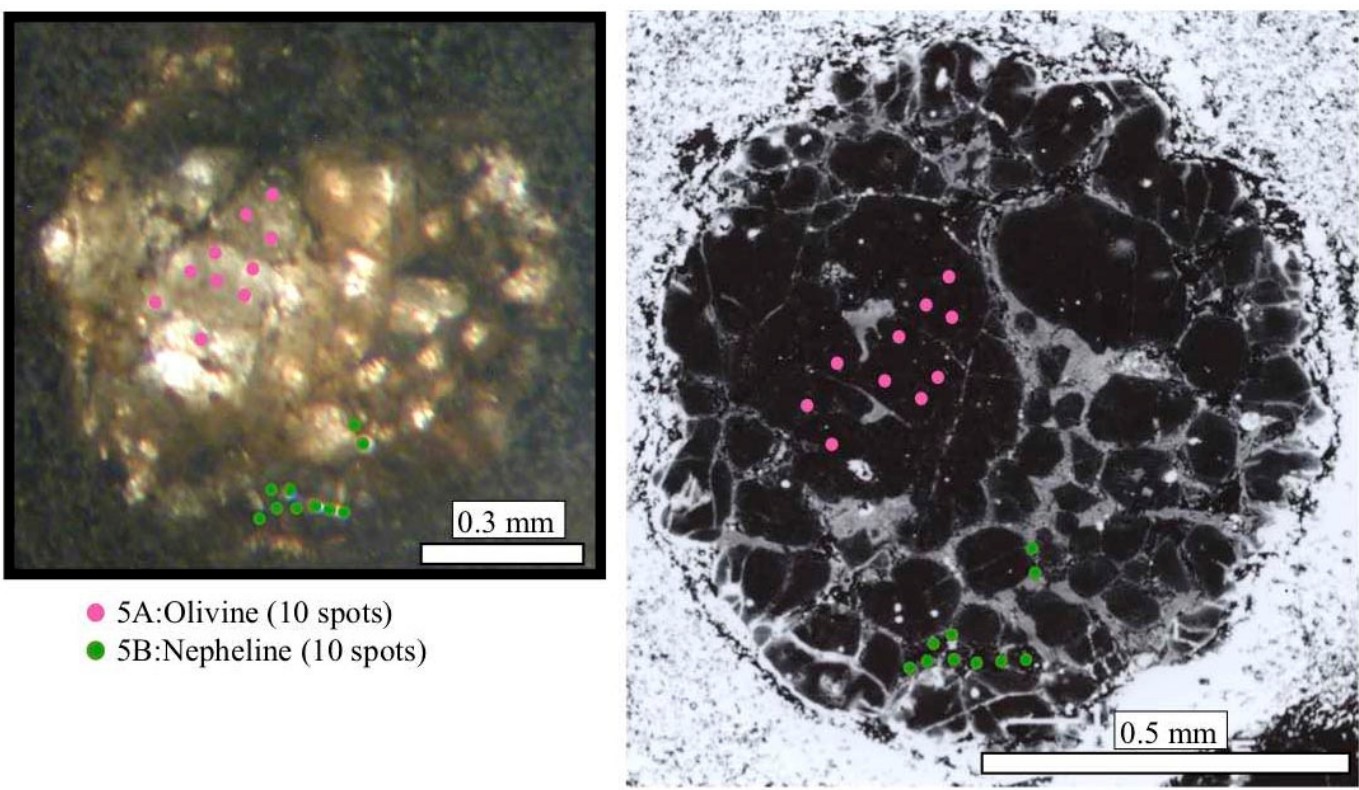

**Figure 7.** Optical view and BSE image of POC-6 showing laser shot locations. The number of laser spots is ten shots for the 5A olivine phase (pink) and for the 5B nepheline phase (green).

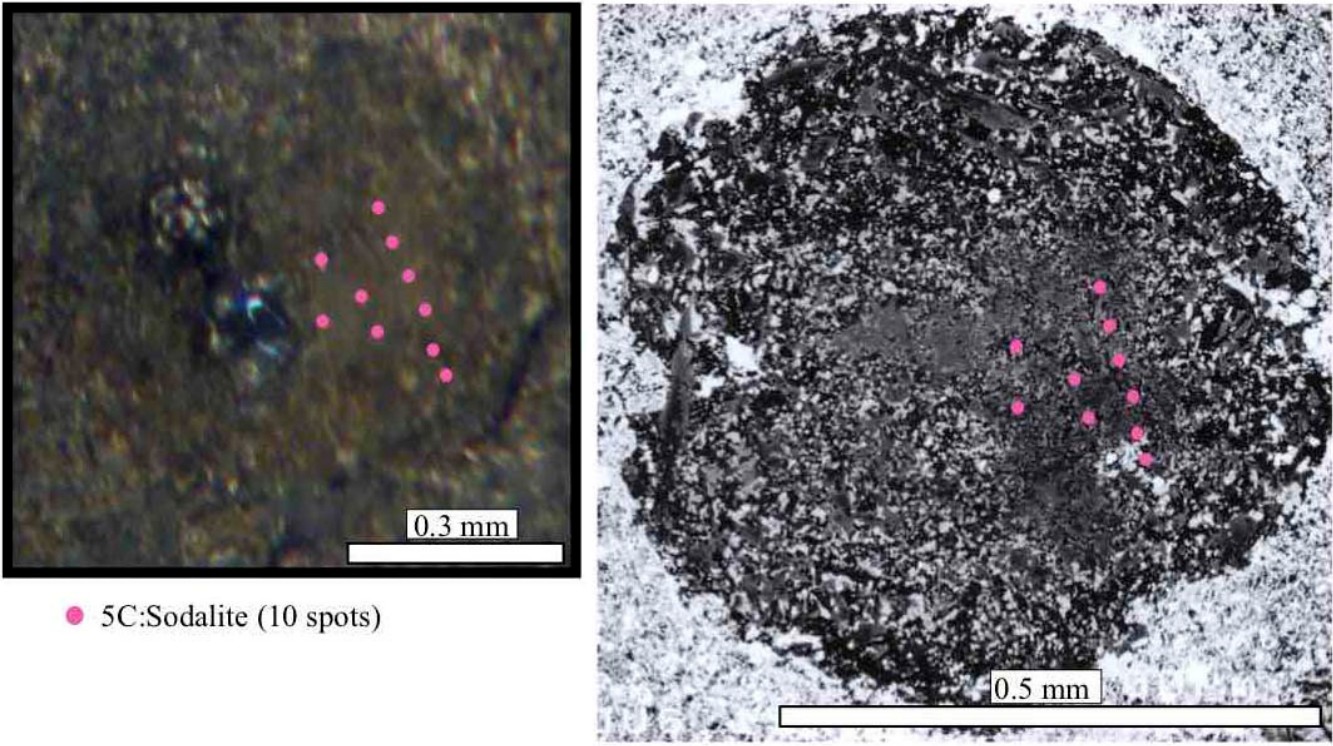

**Figure 8.** Optical view and BSE image of UC-02 showing the laser shot locations. The number of laser spots is ten shots for the 5C sodalite phase (pink).

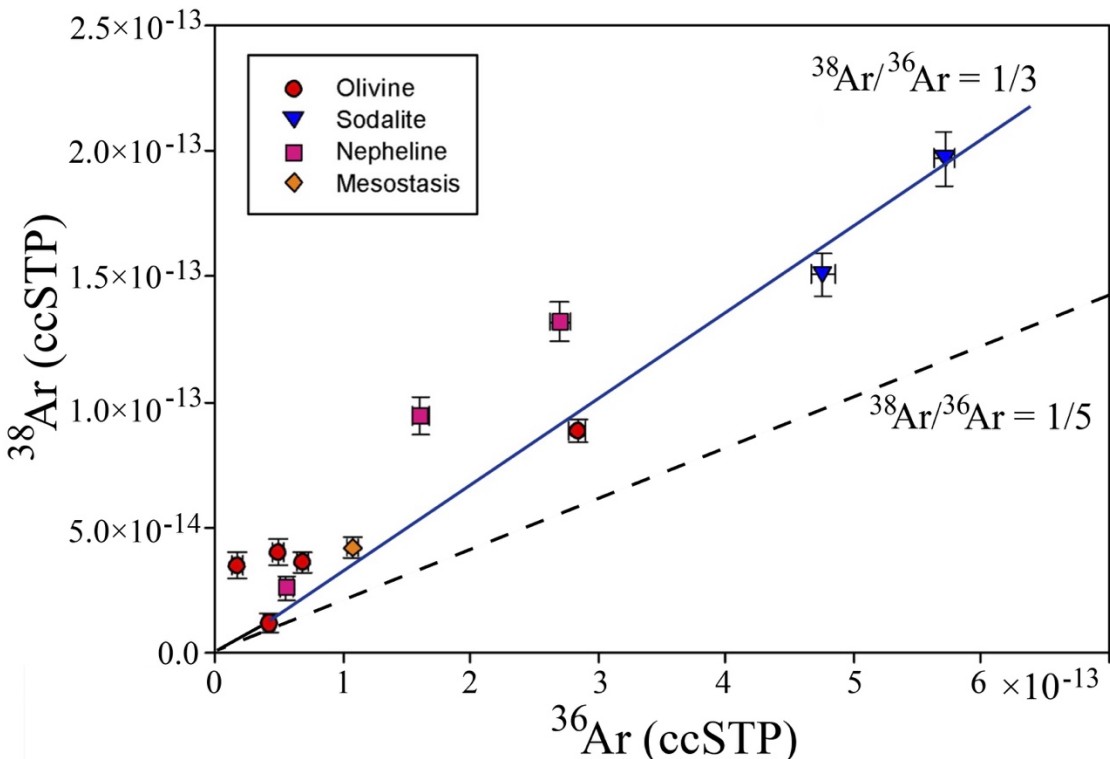

**Figure 9.** $^{38}$Ar versus $^{36}$Ar plot of olivine (circle), sodalite (triangle), nepheline (rectangle), and mesostasis (diamond). The solid and dashed lines in this figure are $^{38}$Ar/$^{36}$Ar = 1/3 and $^{38}$Ar/$^{36}$Ar = 1/5, respectively.

*5.3. Laser Step-Heating $^{40}$Ar/$^{39}$Ar Analyses of Bulk Chondrules*

Argon isotope analyses of five bulk chondrules (~0.5 mm in size) were conducted with the laser step-heating method. Each chondrule was placed in a 2 mm drill hole on an aluminum disk together with age standard grains of 3 gr hornblende [45] and calcium (CaSi$_2$) and potassium (synthetic KAlSi$_3$O$_8$ glass) salts for Ca and K corrections, respectively. Neutron irradiation of the five chondrules was conducted in the core of a 5 MW Research Reactor at Kyoto University (KUR) for 5 h using the hydraulic rabbit facility (sample-capsule-transferring system with hydraulic pressure). The fast neutron flux density is $3.9 \times 10^{13}$ n/cm$^2$/s and is confirmed to be uniform in the dimension of the sample holder ($\varphi16 \times 15$ mm), as little variation in J values of the evenly spaced age standards was observed [46]. The averaged J value and potassium and calcium correction factors are J = 0.00401 ± 0.00003, (40/39) $_K$ = 0.0186 ± 0.0035, (36/37) $_{Ca}$ = 0.000304 ± 0.000019, and (39/37) $_{Ca}$ = 0.00150 ± 0.00009, respectively. Each chondrule was analyzed by a step-heating technique using a 5 W continuous argon-ion laser. During the analyses, the temperatures of the samples were monitored by an infrared thermometer with a precision of 3 degrees within an area of 0.3 mm diameter [47]. The extracted sample gas was analyzed using the mass spectrometer mentioned above. Mass discrimination was checked with atmospheric argon each day. The blanks of argon isotopes were checked several times in each sample analysis because the blank increases during the step-heating analyses. The results are shown in Supplementary Table S2 (ST2) where we can see the blank change during the step-heating analyses. The age spectra and $^{37}$Ar$_{Ca}$/$^{39}$Ar$_K$ ratios are shown in Figure 10. Their integrated ages are 2.77 ± 0.03, 2.78 ± 0.08, 3.07 ± 0.04, 3.18 ± 0.06, and 3.22 ± 0.06 Ga; all errors represent 1σ.

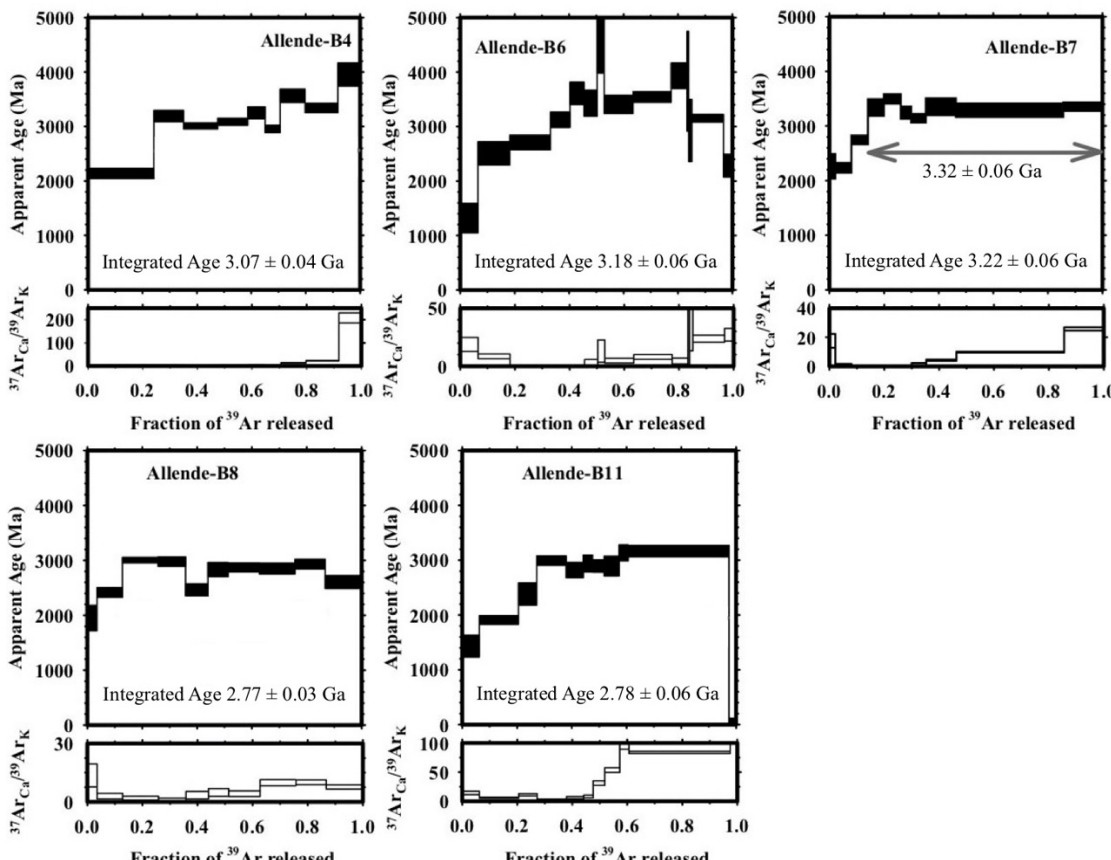

**Figure 10.** Age spectra and $^{37}\text{Ar}_{\text{Ca}}/^{39}\text{Ar}_{\text{K}}$ ratios of bulk chondrules by laser step-heating analyses.

In the conventional stepwise heating analyses of the chondrule with neutron irradiation in the reactor, the cosmogenic $^{39}\text{Ar}$ and the $^{39}\text{Ar}$ formed in the reactor are measured together, underestimating the ages. However, the underestimation is limited because the cosmogenic $^{39}\text{Ar}$ is 1/100 or lower of the $^{39}\text{Ar}$ formed in the reactor. This is because, as described in the discussion, the cosmogenic $^{39}\text{Ar}$ reaches a steady state without increasing indefinitely.

## 6. Discussion

### 6.1. $^{40}\text{Ar}/^{39}\text{Ar}$ Age Determination Method Using the Cosmogenic $^{39}\text{Ar}$

In space, high-energy cosmic ray irradiation can cause a $^{39}\text{K}$ (n, p) $^{39}\text{Ar}$ reaction in extraterrestrial materials before reaching the surface of Earth [1]. Itaya and Hyodo (2022) [1] conducted numerical analyses of the cosmogenic argon formation process and proposed the following equation.

$$N_{39\text{Ar}}\,(t) = -R_{39}N_0 \exp[-(\lambda_{39} + R_{39})t]/(\lambda_{39} + R_{39}) + R_{39}N_0/(\lambda_{39} + R_{39}) \tag{3}$$

Here, $N_{39\text{Ar}}$ is the number of $^{39}\text{Ar}$ atoms. $R_{39}$ and $\lambda_{39}$ are the production rate and the decay constant (0.00258 $\text{y}^{-1}$) of $^{39}\text{Ar}$, respectively. $N_0$ is the total number of atoms of $^{39}\text{K}$ and $^{39}\text{Ar}$. If the production rate of $^{39}\text{Ar}$ is constant under a uniform cosmic ray exposure, and if a long enough period elapses, $N_{39\text{Ar}}\,(t)$ reaches equilibrium as $R_{39}N_0/(\lambda_{39} + R_{39})$ because the first term of Equation (3) approaches zero. This means that the equilibrated amount of $^{39}\text{Ar}$ is proportional to the initial potassium content $N_0$. The cosmogenic $^{39}\text{Ar}$ is also formed by the reaction $^{42}\text{Ca}$ (n, $\alpha$) $^{39}\text{Ar}$. In this case, the equilibrated amount of $^{39}\text{Ar}$ is proportional to the initial calcium content. $^{42}\text{Ca}$ is ~0.6% in Ca, though $^{39}\text{K}$ is ~93% in K, suggesting the effect of Ca-derived $^{39}\text{Ar}$ is limited in the minerals with a low Ca/K ratio.

If we have planetary samples that have experienced high-energy cosmic ray irradiation, and if we can determine the K–Ar age and $^{40}$Ar/$^{39}$Ar ratio of a sample as a reference, the $^{40}$Ar/$^{39}$Ar method can be applied to solve the remaining unknowns. This logic can also apply to date lunar surface rocks by in situ $^{40}$Ar/$^{39}$Ar analyses with equipment mounted on the lunar explorer that has information on the K–Ar age and $^{40}$Ar/$^{39}$Ar ratio of the lunar samples collected by the Apollo mission. The $^{40}$Ar/$^{39}$Ar age can be calculated using Equation (1) described before. The J value in Equation (1) is obtained from the K–Ar age ($t_s$) and the $^{40}$Ar/$^{39}$Ar ratio of a lunar sample collected by the Apollo mission (e.g., [48]) using Equation (2). Since there is no atmospheric contamination on the lunar surface, this application would be easier than that on the Earth's surface. On the other hand, the surface rocks contain the $^{36}$Ar implanted by the solar wind, inducing underestimation of $^{40}$Ar/$^{36}$Ar. To minimize this underestimation, the experimental preheating technique would be necessary. The only drawback is that this approach cannot be applied to samples at different depths from the surface and/or samples with different irradiation histories.

As shown in Table 2, a significant amount of cosmogenic $^{39}$Ar was formed by a $^{39}$K (n, p) $^{39}$Ar reaction in the chondrule-forming materials in the Allende meteorite. The analyzed chondrules would have experienced high-energy cosmic ray exposure with the same irradiation history; at least a centimeter-sized portion (Figure 1) of the Allende meteorite should share the same irradiation history after the disruption of the parent body that took place at ~5 Ma estimated from $^3$He cosmic ray exposure ages of Allende chondrules by Miura et al. (2014) [14]. If the J value is obtained from the isotopic data of one mineral as a reference, the ages of other minerals can be calculated using the analyzed $^{40}$Ar/$^{39}$Ar ratio and the J value obtained. Considering the chemical compositions of minerals found in the Allende meteorite, nepheline is the most appropriate mineral as a reference because of its high abundance of potassium. On the other hand, sodalite is not ideal, because it contains a significant amount of chlorine, some of which might have been converted to Ar by cosmogenic reactions: $^{35}$Cl (n, γ) $^{36}$Cl, β$^-$ decay to $^{36}$Ar, and $^{37}$Cl (n, γ) $^{38}$Cl, β$^-$ decay to $^{38}$Ar. When nepheline is used as an age standard to obtain the $^{40}$Ar/$^{39}$Ar age of minerals, its K–Ar age need to be known. Unfortunately, the nepheline formed by the aqueous alteration in the CV3 parent body of the early solar system is so small in size (see Figure 3), making the K–Ar determination difficult, if not impossible. Therefore, we process the age calculation assuming that the formation age of nepheline is the K–Ar age. The formation age is approximately 4.563 Ga, which is the time of the aqueous alteration in the CV3 parent body of the early solar system, as described above. The age is within the K–Ar age variation (4.26–4.63 Ga) of the Allende chondrules by Jessberger et al. (1980) [49].

Table 2 also shows the J values calculated using the age of 4.563 Ga and the $^{40}$Ar/$^{39}$Ar ratio corrected for the decay of $^{39}$Ar during the term (35 years) from the fall in 1969 to the isotope analyses in 2004 using the following equation.

$$(^{40}\text{Ar}/^{39}\text{Ar})_c = (^{40}\text{Ar}/^{39}\text{Ar})_a e^{\lambda t} \quad (\lambda = 0.00258/\text{y, t} = 35) \tag{4}$$

where $(^{40}\text{Ar}/^{39}\text{Ar})_c$ and $(^{40}\text{Ar}/^{39}\text{Ar})_a$ are equivalent to those in Table 2, respectively.

The apparent ages of chondrule-forming materials were calculated using the J values of three nephelines of BOC-06, POC-13, and POC-06 chondrules (Table 2). Sodalite in the POC-13 chondrule gives 2.1 Ga, much younger than the inferred standard age of nepheline in the same chondrule. These sodalites also yield young ages of 3.7 and 0.8 Ga when nephelines in POC-06 and BOC-06 chondrules are used as the reference values, respectively. Olivine in POC-06 also yields a much younger apparent age (0.8 Ga) than nepheline. On the other hand, nepheline in BOC-06 is much older (7.0 and 8.9 Ga) than the inferred standard age of nephelines in the POC-13 and POC-06 chondrules. As shown in the results, there are no consistent age relationships among the three nephelines. This would be due to the fact that nepheline is characterized by a low closure temperature of the K-Ar isotopic system. The later thermal effects described in the next section might affect significantly the argon loss in nepheline, resulting in no consistent relationship among the three types of chondrules.

Sodalite in POC-13 yields a significantly younger age (2.1 Ga) than that (3.5 Ga) in UC-02. When the J value of nepheline in POC-13 is used for a calculation, olivine in BOC-06 gives ~1.5–5.6 Ga. When the J values of nepheline in POC-06 and BOC-06 are used for calculations, the olivine gives ~2.9–7.5 Ga and ~0.5–3.3 Ga, respectively. Although the age variations seem to be too large, the age calculations were performed without any correction for the incorporation of primitive argon; thus far, no correction method is available. Nevertheless, even if the original argon amount is corrected using the initial argon isotopic ratio from 1 to 295.5, no meaningful change is expected. This wide age variation in the analyzed chondrule-forming materials would be due to the fact that the argon loss took place heterogeneously within the slab (50 mm × 45 mm × 5 mm) of the Allende CV3 meteorite as described in the next section. The heterogeneous Ar loss among the bulk chondrules was observed from the variation of the integrated $^{40}Ar/^{39}Ar$ ages (2.7 to 3.2 Ga) in the $^{40}Ar/^{39}Ar$ analyses. The wide age variation in the analyzed chondrule-forming materials suggests that isotopic heterogeneity among minerals was even greater.

*6.2. K-Ar ($^{40}Ar/^{39}Ar$) Ages of Allende Meteorites*

Jessberger et al. (1980) [49] provided an $^{40}Ar/^{39}Ar$ age of 4.56 ± 0.05 Ga on the Allende chondrule. They also summarized existing K–Ar ($^{40}Ar/^{39}Ar$) ages of Allende meteorites [50–52]. The Allende whole-rock yields a K–Ar age of 4.43 ± 0.09 Ga and an $^{40}Ar/^{39}Ar$ age of 4.57 ± 0.03 Ga. In contrast, the Allende matrix gives a K–Ar age of 3.80 ± 0.09 Ga, which is much younger than the whole-rock K–Ar age. Jessberger et al. (1980) [49] considered that the K–Ar clock in the Allende matrix was probably reset at 3.8 Ga. On the other hand, the chondrules provide K–Ar ages of 4.26–4.63 Ga. These age results suggest that the Allende meteorites have experienced a complex thermal history with argon diffusion after the aqueous alteration at ~4.5 Ga in the CV3 parent body of the early solar system. Huss et al. (2006) [53] described the thermal history on the basis of mineralogical data. Mimura et al. (2020) [54] examined the thermal history of the Allende meteorites by gradual and stepwise pyrolyses of insoluble organic matter and proposed that the meteorites experienced thermal metamorphism at 550–590 °C, followed by an alteration below 300 °C. White inclusions (CAIs) give K–Ar ages of 4.20–5.54 Ga and $^{40}Ar/^{39}Ar$ ages of 4.47–5.43 Ga. In particular, some coarse-grained objects show presolar K–Ar and $^{40}Ar/^{39}Ar$ ages. The presolar $^{40}Ar/^{39}Ar$ ages observed in coarse-grained CAIs would be attributed to $^{39}Ar$ recoil losses during neutron irradiation in a reactor, as discussed later. The meaning of presolar K–Ar ages observed in coarse-grained CAIs is currently unknown.

Figure 10 shows discordant age spectra with variable $^{37}Ar_{Ca}/^{39}Ar_K$ ratios in the step-heating analyses. However, the B7 chondrule shows a plateau age of 3.32 ± 0.06 Ga from 81% of the total $^{39}Ar$ fraction. The variable $^{37}Ar_{Ca}/^{39}Ar_K$ ratios in the plateau suggest that the chondrule experienced a 3.3 Ga event of isotopic homogenization. This isotopic homogenization age is significantly younger than the K–Ar age (3.8 Ga) of the Allende matrix mentioned above. The age spectra also show that the low temperature fractions give younger apparent ages. These younger ages cannot be explained due to the $^{39}Ar$ recoil during the neutron irradiation in the reactor because the $^{39}Ar$ recoil gives the older ages (see the age calculation in Equation (1): the $^{39}Ar$ recoil makes the $^{40}Ar/^{39}Ar$ ratio increase). This type of age spectra with the younger ages in the low-temperature fractions would be in general considered to be due to the diffusion process by the thermal disturbance after the main peak event. The different stage of thermal disturbance and its different length gives the significantly different shape of the spectra. As mentioned above, Jessberger et al. (1980) [49] reported an $^{40}Ar/^{39}Ar$ age of 4.56 ± 0.05 Ga from a chondrule that is within the K-Ar ages of 4.26 to 4.63 Ga, suggesting there was no significant $^{39}Ar$ recoil problem, though CAIs had the $^{39}Ar$ recoil. This suggests that the integrated ages (2.77 to 3.22 Ga) from the $^{40}Ar/^{39}Ar$ analyses of five chondrules are comparable to the K-Ar ages. The sample B7 chondrule gives a plateau age of 3.32 ± 0.06 Ga and an integrated age of 3.22 ± 0.06 Ga, which are similar to each other within the analytical error. This is because of

the limited argon loss. The other four chondrules (B4, B6, B8, and B11) have the integrated ages (3.07, 3.18, 2.77, and 2.78 Ga, respectively) significantly younger than the plateau age of B7. These integrated ages would be due to the fact that the later stage of argon loss was significantly larger than the B7 chondrule. This age variation also suggests that the argon loss took place heterogeneously within the slab (50 mm × 45 mm × 5 mm) of the Allende CV3 meteorite. B4 and B6 chondrules have the fractions older than 3.3 Ga in the middle to high temperatures. These old fractions would be due to the component of the stage before the 3.3 Ga event of isotopic homogenization.

### 6.3. On $^{39}$Ar Recoil during High-Energy Cosmic Ray Irradiation

It has been known that the $^{39}$Ar recoil from very fine-grained glauconite aggregates takes place during neutron irradiation in a reactor, giving significantly older $^{40}$Ar/$^{39}$Ar ages since Yanase et al. (1975) [55]. Smith et al. (1993) [56] provided the first successful $^{40}$Ar/$^{39}$Ar ages of glauconite with irradiation in vacuo by using a microampoule technique of encapsulation. Villa et al. (1983) [57] carried out $^{40}$Ar /$^{39}$Ar analyses of coarse-grained inclusions (CAIs) in Allende meteorite with neutron irradiation in a reactor and concluded that the presolar $^{40}$Ar/$^{39}$Ar ages reported by Jessberger et al. (1980) [49] are mainly due to $^{39}$Ar recoil loss during neutron irradiation in a reactor. These results suggest that there is a possibility of $^{39}$Ar recoil loss from the Allende chondrule-forming materials during high-energy cosmic ray irradiation. The neutron-irradiation-induced recoil of $^{39}$Ar generally affects extremely fine-grained materials such as glauconite aggregates because $^{39}$Ar recoil takes place at the domain close to the crystal surface. The fine-grained materials increase the total size of the crystal surface in comparison with the large single crystals per unit weight. The glauconites studied by Smith et al. (1993) [56] lost a large fraction (up to 64%) of the $^{39}$Ar from the $^{39}$K (n, p) $^{39}$Ar reaction. The Allende inclusions (CAIs) by Villa et al. (1983) [57] lost up to 60% of the fraction. Olivine, nepheline, and sodalite in the investigated chondrules seem to be coarse enough (see Figure 3). The recoil phenomenon of cosmogenic $^{39}$Ar is considered to be largely limited for olivine, nepheline, and sodalite analyzed, as even mesostasis in BOC-06 gives no old age (see Table 2). This is supported by the fact that an $^{40}$Ar/$^{39}$Ar age of 4.56 ± 0.05 Ga from a chondrule is within the K-Ar ages of 4.26 to 4.63 Ga [49], suggesting there were no significant $^{39}$Ar recoil problems.

## 7. Summary

The argon isotopic compositions of chondrule-forming minerals of the Allende (CV3) meteorite were examined to evaluate the possibility of in situ $^{40}$Ar/$^{39}$Ar dating of planetary surface rocks based on cosmogenic $^{39}$Ar without neutron irradiation in a reactor. In situ argon isotope analyses on selected chondrule-forming minerals in petrographic sections with a special focus on the spatial distribution of K-bearing nepheline and Cl-bearing sodalite using a Nd:YAG pulse laser confirmed a significant amount of cosmogenic $^{39}$Ar that formed by a $^{39}$K (n, p) $^{39}$Ar reaction in an extraterrestrial environment. This suggests that $^{40}$Ar/$^{39}$Ar dating based on cosmogenic $^{39}$Ar would be applicable to various extraterrestrial rocky materials. Therefore, we described the $^{40}$Ar/$^{39}$Ar age determination method using cosmogenic $^{39}$Ar and carried out the age calculation of chondrule-forming minerals using the J values of nephelines as an inferred standard. The results did not show any consistent relationship among the three types of chondrules, suggesting the isotopic heterogeneity among minerals took place during the heterogeneous argon loss stage after the 3.3 Ga event of isotopic homogenization. To further develop this method, a more systematic assessment using various meteorite samples is required compared to previous studies.

**Supplementary Materials:** The following are available online at https://www.mdpi.com/article/10.3390/min13010031/s1. Supplementary Figure S1: Back-scattered electron (BSE) images of the 37 chondrule objects; Supplementary Figure S2: X-ray compositional maps (CMs) of K-alpha radiation images of eight elements of Si, Al, Mg, Fe, Ca, Na, K and Cl of the objects; Supplementary Table S1: EMP analyses data of mineral phases in the chondrules and CAIs; Supplementary Table S2:

Argon isotope analytical data of bulk chondrules by step-heating method; Supplementary Data (SD): Petrographic description and mineral chemistry of the 37 chondrule objects (including olivine fragments) and CAIs. Ref. [58] is cited in Supplementary Materials file.

**Author Contributions:** Conceptualization, T.I. and H.H.; methodology, H.H., T.T. and C.G.; software, H.H.; EMP and argon isotope analysis, Y.T. and H.H.; resources, T.I. and H.H.; writing—original draft preparation, Y.T., H.H., T.T. and C.G.; writing—review and editing, T.I., T.T. and H.H.; supervision and project administration, T.I. All authors have read and agreed to the published version of the manuscript.

**Funding:** This research received no external funding.

**Data Availability Statement:** The data presented in this study are available in Supplementary Materials.

**Acknowledgments:** The neutron irradiation experiment was carried out in the visiting researcher program in the Kyoto University Reactor (KUR). We thank the staff of Kyoto University, especially K. Takamiya, for his help in the course of the experiment. We sincerely thank Ichiro Kaneoka of the University of Tokyo who carefully read the manuscript and gave us many comments that improved the content of the manuscript.

**Conflicts of Interest:** The authors declare no conflict of interest.

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
