# Peer review of "In Situ Argon Isotope Analyses of Chondrule-Forming Materials in the Allende Meteorite: A Preliminary Study for 40Ar/39Ar Dating Based on Cosmogenic 39Ar"

_minerals, doi:10.3390/min13010031_

Round 1

Reviewer 1 Report (New Reviewer)

The authors have extensive experience in the application of the highly sensitive 40Ar/39Ar dating method in the study of terrestrial and space objects. This article is devoted to the study of chondrules-forming materials in the Allende meteorite. The results of two methodological approaches are compared in the work. This is, on the one hand, conventional dating by the method of stepwise heating using a continuous argon-ion laser, on the other hand, in–situ point dating of minerals that make up chondrules in the meteorite. In the latter variant, the authors record significant amounts of 39Ar argon formed under the influence of cosmic radiation and propose using a mineral with a "known" age (nepheline) as a neutron flux monitor to calibrate the neutron flux.

 The work presents new, interesting materials that meet the highest requirements, and it deserves publication. From my point of view, the following two issues need to be resolved:

1)                 On the one hand, with the help of point dating, the authors record a significant amount of 39Ar formed in space conditions in the minerals of chondrules. But this circumstance is not taken into account by them in any way when processing and interpreting the data of conventional stepwise heating, although it would seem that 39Ar of cosmic origin should lead to a significant underestimation of the measured age of minerals, and differing depending on the minerals degassed at each specific stage.

2)                 When choosing and justifying nepheline as a monitor of the cosmic neutron flux, the authors assign it an age corresponding to that determined by other methods of the aqueous alteration of the meteorite's parent body in the early solar system (4.563 Ga). This contradicts the existing ideas that, firstly, nepheline is characterized by a low closing temperature of the K/Ar isotopic system - no higher than that of feldspar. At the same time, the results obtained by the authors themselves indicate significant late thermal effects that led to a significant rejuvenation of the isotopic K/Ar system of minerals. At least this contradiction should be discussed by the authors.

Author Response

Reviewer 1

Comments and Suggestions for Authors

The authors have extensive experience in the application of the highly sensitive 40Ar/39Ar dating method in the study of terrestrial and space objects. This article is devoted to the study of chondrules-forming materials in the Allende meteorite. The results of two methodological approaches are compared in the work. This is, on the one hand, conventional dating by the method of stepwise heating using a continuous argon-ion laser, on the other hand, in–situ point dating of minerals that make up chondrules in the meteorite. In the latter variant, the authors record significant amounts of 39Ar argon formed under the influence of cosmic radiation and propose using a mineral with a "known" age (nepheline) as a neutron flux monitor to calibrate the neutron flux. 

The work presents new, interesting materials that meet the highest requirements, and it deserves publication. From my point of view, the following two issues need to be resolved:

1) On the one hand, with the help of point dating, the authors record a significant amount of 39Ar formed in space conditions in the minerals of chondrules. But this circumstance is not taken into account by them in any way when processing and interpreting the data of conventional stepwise heating, although it would seem that 39Ar of cosmic origin should lead to a significant underestimation of the measured age of minerals, and differing depending on the minerals degassed at each specific stage.

Response: Yes, it is. In the conventional stepwise heating analyses of the chondrule with neutron irradiation in reactor, the cosmogenic 39Ar and the 39Ar formed in reactor are measured, making the ages underestimated. However, the underestimation is limited because the cosmogenic 39Ar is 1/100 or lower of the 39Ar formed in reactor. This is because, as can be seen from Equation 3, the cosmogenic 39Ar reaches a steady state without increasing indefinitely. The previous researchers have not considered it as they don't understand this issue at all. Or maybe they know it and don't discuss it.

2)When choosing and justifying nepheline as a monitor of the cosmic neutron flux, the authors assign it an age corresponding to that determined by other methods of the aqueous alteration of the meteorite's parent body in the early solar system (4.563 Ga). This contradicts the existing ideas that, firstly, nepheline is characterized by a low closing temperature of the K/Ar isotopic system - no higher than that of feldspar. At the same time, the results obtained by the authors themselves indicate significant late thermal effects that led to a significant rejuvenation of the isotopic K/Ar system of minerals. At least this contradiction should be discussed by the authors.

Response: Yes, nepheline is characterized by a low closure temperature of K-Ar isotopic system. The later thermal effects might affect significantly the argon loss in nepheline, resulting in no consistent relationship among the three types of chondrules. We added the following sentences in discussion (see Section 6-1).

“This would be due to that nepheline is characterized by a low closure temperature of K-Ar isotopic system. The later thermal effects described in the next section might affect significantly the argon loss in nepheline, resulting in noconsistent relationship among the three types of chondrules.”

Submission Date

13 October 2022

Date of this review

27 Oct 2022 16:13:12

Reviewer 2 Report (New Reviewer)

Please see the attached pdf file.

Author Response

Reviewer 2

Comments and Suggestions for Authors

Please see the attached pdf file.

------------------------------------------------------------------------------------------------------
General comment
This manuscript describes the possibility of in-situ 40Ar/39Ar dating methods using cosmogenic 39Ar and discusses the effects of Cl and recoil for the Ar/Ar dating. The contents of this manuscript are in focus on Minerals (and its special issue) and will be helpful for readers. Therefore, I recommend publication. However, several moderate revisions are required to publish this manuscript.
What I am concerned is below; the authors should estimate and mention the effects
of Ca contents to the production of cosmogenic 39Ar. It is not clear for me how much we can apply the in-situ Ar/Ar dating to the extraterrestrial surface rocks (How much does authors think J-values are homogeneous on the lunar surface, in particular around craters? Line41- 42). The interpretation of Ar/Ar ages obtained from Allende bulk chondrules should be reconsidered by comparing previous studies based on mineralogy and/or organic chemistry. Moreover, I think that the authors should better check the recoil effects in chondrites in previous studies treating chondrites rather than terrestrial rocks. Line-by-line comments are described below.

------------------------------------------------------------------------------------------------------
Pages 1-8
Line 18
~ at least three “textural” types of ~ may be better.

Response: We did it.

Line 20 “plagioclase”
If authors are sure this is not glass but crystal and its chemical composition is relatively homogeneous, “anorthite” or “albite” are better description than “plagioclase”. In my opinion, the word “plagioclase” should be used for indicating those with a significant chemical zoning, such as that in volcanic rocks.

Response: We have lots of chemical data (see ST1 and SD). Fig. SD12 (Or-Ab-An diagram) shows 0.78-0.98 of An. This is a reason why we used the word “plagioclase”

Line 41-42
It is not clear for me why in-situ Ar/Ar dating can reduce the uncertainty of lunar crater chronology. As authors mentioned, to conduct in-situ Ar/Ar dating of rocks on a planetary surface, at least one sample with a known age whose irradiation condition is identical to that of unknown samples is needed. Since irradiation conditions may be different between inside and outside of a crater due to impact resetting, the K-Ar or Ar/Ar age of a rock from the inside of crater may be required before in-situ Ar/Ar dating. If so, the in-situ analysis does not seem to be necessary. I think it is better to add a little more explanation how to reduce the errors of crater chronology.

Response: We exchanged a word from “can” to “may”. We added an explanation how to reduce the errors of crater chronology in the discussion section.

Line 49
[~ the investigated “chondrules” contain~] may be correct?

Response: We did it.

Line 50
“of” -> “on” ?
Response: We did it.

Line 51-52
This part is a little hard to understand for me. For my understanding, the presence of cosmogenic 36Ar results in over-correction of trap 40Ar because 40Ar/36Ar is generally used for trap (initial) Ar correction. Is this correct? I think more explanation is needed in this part.

Response: We added some phrase after 36Ar “produced by neutron capture on 35Cl.”

Line 57

“mineralogical reconnaissance” may not be a common word. I think just “mineralogical observation” is enough.

Response: We did it.

Line 58-60
I totally agree with this part.

Response: Thanks

Line 74
As authors may know, Allende also contains AOAs which looks like CAIs. If authors can tell CAIs from AOAs in Fig. 1, it may be better to point them out to make sample description look more convincing.

Response: We changed the sentence to below in sample description.

“The slab contains abundant chondrules and some Ca-Al-rich inclusions (CAIs) (Fig. 1) and may also contain the amoeboid olivine aggregates (AOAs) studied by Grossman and Steele (1976).”

Line 69
Planetary surface rocks generally contain abundant 36Ar implanted by the solar wind. Such 36Ar may also induce underestimation of 40Ar as mentioned in Line 51-52, in particular planetary surface volcanic rocks with a potential high trap 40Ar/36Ar. If considering the planetary surface rocks, I think the authors should discuss the effects of solar wind-derived 36Ar somewhere.

Response: Yes, we put the following sentence in the discussion (see Section 6-1).

“On the other hand, the surface rocks contain the 36Ar implanted by the solar wind, inducing underestimation of 40Ar. To minimize underestimating, the preheating experimental technique would be necessary.”

Line 151 (Fig. 2)
I think X-ray maps of Ca or Al may be better to add to show where anorthite is (Fe may not be necessary?).

Response: We exchanged Fe to Ca map in Figure 2 according to your comment.

Line 251-252
I am wondering if there are any effects of Ca because olivine in Allende contains relatively high amounts of Ca (~0.6 wt%). As authors mentioned in method section, Ca can capture neutrons and generate the interference 39Ar (42Ca(n,α)39Ar). Although the amount of 42Ca is very small, some silicates contain Ca 2000~3000 times higher than K (data in ST1) and Ca may have a potential to be one of the cosmogenic 39Ar sources. I feel it is better for authors to discuss the effects of Ca (and possibly Fe?) throughout the manuscript, even if it does not change the results.

Response: Thank you for pointing out the effect of Ca. We put the following sentences in this part.

“On the other hand, olivine has significant amount of CaO up to 0.8wt.% (see ST1). This means the cosmogenic 39Ar may be in part formed by the reaction of 42Ca(n,α)39Ar”

Line 261
Who are “they”? Miura et al. (2014)?
Response: Miura and Nagao of them are noble gas geochemists of University of Tokyo and have published many papers on meteorites. Kimura of them is a mineralogist who is familiar in meteorites.

Line 261-262
I feel there is a small contradiction in this part and the explanation of Fig. 9.
Authors explained that 38Ar/36Ar plots on the line with a slope of 1/3 (Fig. 9) are due to Cl derived Ar because the natural ratio of Cl is 37Cl/35Cl = 1/3 (0.33). On the other hand, authors said the low 38Ar/36Ar ratio (< 0.19) is due to the addition of Cl-derived 36Ar in this part.

Response: Thanks for pointing out a contradiction for the reason to explain 38Ar/36Ar ratio of sodalite. We have misunderstood the reason. We revised this part to the following sentences.

“Since the extracted gas was purified, this would not be due to the formation of hydrogen chlorides during the isotope analyses. The cosmogenic reactions of 35Cl (n, γ) 36Cl, β- decay to 36Ar and 37Cl (n, γ) 38Cl, β- decay to 38Ar give the ratio much smaller than the natural ratio of 37Cl/35Cl =1/3 because of difference of the thermal neutron cross sections for 37Cl and 35Cl [44]. The sodalite has 0.3 -2.8 wt.% of CaO (see ST1), suggesting that the cosmogenic 36Ar formed by the reaction 40Ca (n, n?) 36Ar. This makes the ratio much lower. On the other hand, the cosmogenic 38Ar can form by the reactions 42Ca (n, n?) 38Ar and 41K (n?, ?-) 38Ar.”

I think neutron cross sections for 35Cl and 37Cl (or some other parameters) may be different, and the ratio of Cl-derived 38Ar/36Ar will be much smaller than the ratio of natural 37Cl/35Cl ratio (1/3) (cf. the thermal neutron cross sections for 35Cl and 37Cl are 44 and 0.43 barns, respectively [ref. 44]). More explanation may be needed in this part and Fig. 9. Pages 9-18

Response: see above response.

P12 Line 23-27
Please also see comments above (for Line 261-262).
I think a plot showing 38Ar/36Ar versus Ca contents is useful to visualize the presence of Cl derived Ar. Since cosmogenic Ar is mainly derived Ca, 38Ar/36Ar will be proportional to Ca contents. If there are some plots out of the proportional line, there should be any Ar sources other than trap or Ca. However, I am not sure if this graph is useful in this study, and this is just a comment.

Response: see above response.

P14 Line 55-90
I was a little confused in this part.

The authors suggest an isotopic homogenizing event at 3.3 Ga. On the other hand, the authors also indicate that “the argon loss took place heterogeneously”. How to achieve this situation? Moreover, I am not sure what kind of events can induce such homogenization. The thermal history of Allende is estimated by many papers based on not only gases but also minerals and organic matter. For example, “the Allende meteorite experienced thermal metamorphism at 550‒590°C followed by an alteration below 300 °C.” (Mimura et al. 2020) Such alteration event may have occurred at ~4.5 Ga. If this is the case, the event at 3.3 Ga should be below 300 °C and I do not think such low temperature event can homogenize Ar by diffusion in high-temperature minerals such as olivine. Authors need to discuss consistency between the event at 3.3 Ga and other thermal histories estimated in previous paper (including artificial effects in obtained ages). It is also good way to indicate semi-quantitative peak temperature and/or heating durations which is needed for Ar homogenization by diffusion.

Anyway, since this section is for results, most parts written here should be moved to the discussion part (6.3?).

Response: We moved this part to the discussion part (see Section 6-2) and discussed more in detail.

P14 Line 102-103
I think authors should mention about the Ca effects if samples have high Ca/K ratios. (Please see the above comments)

Response: We put the following sentences here.

“The cosmogenic 39Ar is also formed by the reaction 42Ca (n, ?) 39Ar. In this case, the equilibrated amount of 39Ar is proportional to the initial Calcium content. 42Ca is ~0.6% in Ca though 39K is ~93% in K, suggesting the effect of Ca derived 39Ar is limited in the minerals with low Ca/K ratio.”

Line 109-111
I would like to know how far J-values will be the same around craters on the lunar surface.

Response: We would like to know that too. It may be possible to solve it by repeating the experiment. As a result, we believe that the error will be smaller.

Line 111-112
Although there is no atmospheric contamination, it is not clear if there is a trap Ar in unknown samples. There is also SW-derived Ar on lunar surface. I feel that this application must be useful and worth developing but would not be so easier than that on the Earthʼs surface.

Response: Yes, the surface rocks contain the 36Ar implanted by the solar wind, inducing underestimation of 40Ar. To minimize underestimating, the preheating experimental technique would be necessary as described above.

Line 124
“identical” -> “ideal”?

Response: We did it.

Line 148~ and 6.2
For my understanding, when a whole rock sample is irradiated, the recoil effect can induce older 40Ar/39Ar ages for K-richer phase and younger ages for K-poorer ones because neutron-induced 39Ar is moved (~0.1 μm) from K-richer phase surface to K-poorer ones in total (e.g., Dixon et al. 2003) (part of 39Ar may be lost from the sample). Recoil in reactor is relatively common in chondrites because its grain size is small, resulting in apparent older/younger ages in low-/high- temperature fractions in stepwise heating. Therefore, the young ages of olivine in this study (in particular BOC where K-poor olivine is adjacent to Krich phases) may be partly attributed to the recoil effect. Although the authors described that nepheline and sodalite are coarse enough (Line 179), individual grains look very small based on Fig. 3.

Response: The argon gas diffuses out along the interface between the crystals. When argon gas jumps out by the recoil from a certain crystal to the interface, it diffuses out along the interface. It is unlikely that the argon gas in the interface will enter other crystals that are in contact. It must be noted that olivine is in contact with mesostasis through an interface.

Supplementary files
SD
There are some typo and mistakes, please check them.

“chondorules” -> “chondrules”
Figure captions: “Enlarged images” -> “An enlarged image”
“olivine fragments shows” -> “olivine fragments show”
Pyroxene with low-Ca is described as “low-Ca pyroxene” in the manuscript, while it is written as “orthopyroxene” in SD. If authors did not check the crystal structure of pyroxene, authors should unify the description as “low-Ca pyroxene” since there is low-Ca (<Wo5) clinopyroxene in meteorites.

Response: We checked SD and revised the points you made.

SF1
Some scale-bars added by authors look different from those originally added by SEM, such as in BOC-02, BOC-10, POC-04...etc. Are these scales correct? Authors must check them all.
Cf. BOC-01 looks 1.6mm in diameter in elemental map in SF2, while it looks <1.2 mm in diameter in SF1.

Response: We checked then all and revised the error scales.

ST1
Some data seem to be under a detection limit (not detected). Authors should check it and must rewrite 0.00 to “n.d.” if the data is below the detection limit. I think the symbol “-” means “not measured”, if so, authors should better note it.

Response: We checked ST1 and revised the points you made.

ST2
According to 39Ar/36Ar versus 40Ar/36Ar graph (isochron plot), there seems to be a trap Ar in each analysis. In my eyes, the 0615B6 analysis may have relatively high trap 40Ar/36Ar whose origin is unknown. Although the presence of trap Ar makes the discussion more complex, there should be some meanings. (this is just a comment)

Response: Thanks for your comments. We would like to examine on the trap Ar in future.

Submission Date

13 October 2022

Date of this review

28 Oct 2022 06:46:53

Reviewer 3 Report (New Reviewer)

I found this manuscript a little challenging to follow however I am not affluent in this particular technique and the calculations required around it. I feel that further additions would be beneficial, such as adding in more context to the goals in the introductory sections so your reader can easily understand the motivations around this study. There are also some elements of the results which should be in the discussion sections. 

The discussion would benefit from more context and evaluation of what the results mean in the context of other dating studies- for example, is the variation you detected real, is it comparable to other dating studies, what does it mean for the CV parent body etc. Below I outline some specific points that I picked up on whilst reading:

Table 1- What do the symbols mean? Specifically the ?

Fig 2- Recommend adding titles to each row for ease of understanding, please make all BSE maps the same size, arrow to say 'increasing concentration'.

Fig 3- Boxed area in fig 2 not very obvious- I recommend adding a close up of this to figure 3/giving more context. 3b- where is this in part a? Recommend adding labels to 3b so the reader understands what they are looking at. E.g., what are the lines? what are the colors?

L159- rim part= chondrule rim, core part= chondrule core

L224-226- I do not understand what this sentence is saying

Figs 4+5- What do you mean by optical view? Are these plane polarized images? Laser shot= laser spot?

P13 L30- What is the reasoning behind this statement?

P14 L55-90- These are discussion points rather than results

P14 L114- How do you define what the radiation histories would be? Do you mean you wouldn't be able to define the age if two samples on the same planet have different radiation histories? This feels like a big setback to the methodology, and should be discussed further.

P15 L118- Why? What is the reasoning here?

P15 L148- How do you explain the lack of age relationships? What does it mean?

P15 L166- Since= compared to?

P16- I struggled to follow the logic of there being no Ar recoil problem- consider rephrasing including reasoning as to why it doesn't appear to be an issue.

Discussion needs some additional work- I found it hard to follow the reasoning and the order didn't appear to be as logical as it could be. Needs robust discussion of your methods putting them in clear context with previous work. How does it compare to ages from other methods? Any room for improvement? Could there be any errors in your work?

Overall this is should be published after careful consideration of the messages the authors are conveying. With some reframing, rephrasing and potential reordering of the discussion/results sections, I feel it would be greatly improved and easier for the reader to follow and understand. 

Author Response

Reviewer 3

Comments and Suggestions for Authors

I found this manuscript a little challenging to follow however I am not affluent in this particular technique and the calculations required around it. I feel that further additions would be beneficial, such as adding in more context to the goals in the introductory sections so your reader can easily understand the motivations around this study. There are also some elements of the results which should be in the discussion sections. 

The discussion would benefit from more context and evaluation of what the results mean in the context of other dating studies- for example, is the variation you detected real, is it comparable to other dating studies, what does it mean for the CV parent body etc. Below I outline some specific points that I picked up on whilst reading:

Table 1- What do the symbols mean? Specifically the ?

Response: Circle indicates presence. The mark “?” indicates that there may be. No mark indicates no presence.

Fig 2- Recommend adding titles to each row for ease of understanding, please make all BSE maps the same size, arrow to say 'increasing concentration'.

Response: We put the sample number on each map. The first author uses a diagram created during her graduate student days. We don't have the original drawing, so We can't make it the same size. We are sorry. Please forgive me for this.

Fig 3- Boxed area in fig 2 not very obvious- I recommend adding a close up of this to figure 3/giving more context. 3b- where is this in part a? Recommend adding labels to 3b so the reader understands what they are looking at. E.g., what are the lines? what are the colors?

Response: We revised Fig. 3 according to your comments.

L159- rim part= chondrule rim, core part= chondrule core

Response: Yes, it is so.

L224-226- I do not understand what this sentence is saying

Response: We simplified this sentence as seen below.

“The sample extraction-purification line and the detector of argon analyses were improved for the 40Ar/39Ar dating of submillimetre-sized single mineral grain.”

Figs 4+5- What do you mean by optical view? Are these plane polarized images? Laser shot= laser spot?

Response: An image captured by a CCD camera of light transmitted from below the thin section.

P13 L30- What is the reasoning behind this statement?

Response: We wanted to say that there are various origins of argon isotopes.

P14 L55-90- These are discussion points rather than results

We moved this part to the discussion part and discussed more in detail.

P14 L114- How do you define what the radiation histories would be? Do you mean you wouldn't be able to define the age if two samples on the same planet have different radiation histories? This feels like a big setback to the methodology, and should be discussed further.

Response: It's not a setback in methodology. We would like to say that special care must be taken in sample collection because correct results cannot be obtained in such cases.

P15 L118- Why? What is the reasoning here?

Response: This is because it seems that the same irradiation conditions can be obtained with this size from the experience of neutron irradiation in nuclear reactors.

P15 L148- How do you explain the lack of age relationships? What does it mean?

Response: We thought the methodology was correct and tried to calculate, but the results did not match among the three kinds of nephelines. This is because the argon diffusion occurred heterogeneously due to the complicated thermal history, which will be described in a new section (6.2) of discussion.

P15 L166- Since= compared to?

Response: Meaning “since Yanase's paper”.

P16- I struggled to follow the logic of there being no Ar recoil problem- consider rephrasing including reasoning as to why it doesn't appear to be an issue.

Discussion needs some additional work- I found it hard to follow the reasoning and the order didn't appear to be as logical as it could be. Needs robust discussion of your methods putting them in clear context with previous work. How does it compare to ages from other methods? Any room for improvement? Could there be any errors in your work?

Response: We changed the discussion significantly. We compared our dating results with reviews of meteorite dating studies. It makes sense.

Overall this is should be published after careful consideration of the messages the authors are conveying. With some reframing, rephrasing and potential reordering of the discussion/results sections, I feel it would be greatly improved and easier for the reader to follow and understand. 

Response: We improved the discussion significantly with a new section (6.2).

Submission Date

13 October 2022

Date of this review

27 Oct 2022 09:51:58

Round 2

Reviewer 2 Report (New Reviewer)

The revised manuscript describes the possibility of in-situ 40Ar/39Ar dating methods using cosmogenic 39Ar and discusses the effects of Cl and recoil for the Ar/Ar dating. The contents of this manuscript are in focus on Minerals (and its special issue) and will be helpful for readers. I think the authors dealt with most of the previous reviewers comments. Therefore, I recommend publication. However, I found some mistakes and recommend double-checking the text.

For example, I think there must be space between the number and its units.

Line 147, 4.5_Ga

Line 151 550-590_oC, 300_oC

…etc.

Please check them.

Just a comment (below)

Authors may think that recoiled 39Ar near the crystal surface must diffuse out and a crystal may lose 39Ar by recoil effect. I think this is true for crushed (powdered) samples or fragments made up of one type of mineral (e.g., only plagioclase). On the other hand, a recoil effect in fragments with several types of fine-grained minerals, such as chondrites, may be more complicated. Please refer to Dixon et al (2003) MaPS, 38, Nr 3, 341-355.

Author Response

Comments and Suggestions for Authors

The revised manuscript describes the possibility of in-situ 40Ar/39Ar dating methods using cosmogenic 39Ar and discusses the effects of Cl and recoil for the Ar/Ar dating. The contents of this manuscript are in focus on Minerals (and its special issue) and will be helpful for readers. I think the authors dealt with most of the previous reviewers’ comments. Therefore, I recommend publication. However, I found some mistakes and recommend double-checking the text.

For example, I think there must be space between the number and its units.

Line 147, 4.5_Ga

Line 151 550-590_oC, 300_oC

…etc.

Please check them.
Response: We did them.

Just a comment (below)

Authors may think that recoiled 39Ar near the crystal surface must diffuse out and a crystal may lose 39Ar by recoil effect. I think this is true for crushed (powdered) samples or fragments made up of one type of mineral (e.g., only plagioclase). On the other hand, a recoil effect in fragments with several types of fine-grained minerals, such as chondrites, may be more complicated. Please refer to Dixon et al (2003) MaPS, 38, Nr 3, 341-355.

Response: We thank you sincerely.

Submission Date

13 October 2022

Date of this review

16 Dec 2022 07:53:31

This manuscript is a resubmission of an earlier submission. The following is a list of the peer review reports and author responses from that submission.

Round 1

Reviewer 1 Report

I note that the title of the MS has been changed,which sounds more appropriate to the text.

I have no further  questions and concerns.

Reviewer 3 Report

General comments

The manuscript is interesting and demonstrated the use of K-Ar system to determine the ages of chondrule minerals. However, there are several issues which need to be addressed before accepting it for further consideration. The chondrules are irradiated by cosmic rays in Allende meteorite, irradiation also produces other Ar isotopes. It is not clear if these effects are considered in the calculations and corrections are made for it. The important missing point is, what is the age of the isochron means. Is it the irradiation age or the age of the formation of secondary minerals in the chondrules? Because the K-Ar systematics is mainly dominated by secondary minerals (such as sodalite and nepheline) the isochron age is not likely represent the chondrule formation age. It needs to be revised and stated clearly. Some of the ages of chondrules in Table 2 are unrealistically high (5.57 Ga), which is even older than solar system. It is not correct.

To be honest, I am not the expert to comment on the method section for isotope ratio measurements, therefore the comments from other reviewer who is expert with this technique need to be incorporated before considering further for publication. The conclusion section is missing and discussion is not structured around the most important findings of this work. Therefore, overall I suggest that author needs to improve it with major revision and opinion from expert on the methods need to be asked before proceeding further with this manuscript. 

Specific comments

Without introduction, readers will not understand what is J in the abstract.

47: The referencing style is different from the reference 1. Please check it with journal policies and revise it accordingly.

64-67: The term chondrule-forming minerals is misleading. These are the minerals that are present in chondrule, but they are not necessarily chondrule forming minerals. Please change this wording throughout the manuscript.

77: There are numerous studies on this meteorite: Please give reference to selective studies or delete this sentence.

82-83: The Al-Mg chondrule ages range from 0 to 4 Ma after CAIs. Please check most recent references.

  87: This result constrains the formation (of what?)

98: natural and synthetic silicates and oxides: Could you please mention the exact minerals and synthetic compositions used for the calibration.

187-188: The modal abundance of nepheline and sodalite is variable among the objects. However, their abundances are generally higher in the former than in the latter: The sentences are not coherent. Reader cannot guess what is latter and what is former. Please state it clearly.

259: Chondrule-forming minerals, please change it.

368-369: Nepheline is a secondary mineral; therefore, it cannot be older than the age of meteorite itself. These ages are not consistent with chondrule ages or ages of any solar system object.

362:381: The whole discussion in this paragraph is bit difficult to understand. Also, the calculations are probably not considering the corrections needed to compare the sodalite and nepheline ages. I would suggest if there is no correction method known so far, then this discussion does not make any sense. It is better to shorten this discussion and focus the discussion on the actual ages authors reported in Table. The calculation of J values

Summary and future scope is mixed with discussion and important conclusions of this work are missing. Please add conclusion section to the manuscript to highlight important conclusions of your work.

Round 2

Reviewer 2 Report

See previous reI started to look at the revised paper and at the authors comments.  Up through page 14, there have been no substantive changes to the manuscript.  The responses to my comments indicate to me that the authors do not understand what they are doing, except at a superficial level.  For example, their response to my concern about the correlation in abundances of 36Ar, 38Ar, 39Ar, and 40Ar, three of which are stable (one produced by a long-lived radioactivity) and one of which short lived, is brushed aside with a list of nuclear reactions.  That isn't the point.  Stable nuclides produced by comic rays, regardless of the reaction, will build up linearly with time (assuming a constant cosmic ray flux).  39Ar, with a half life of 269 years, will reach a steady state in roughly 2000 years and its abundance will not increase, unless the flux of cosmic rays increases.  Their data in Table 2 show that all Ar isotopes are correlated.  Why should a nuclide that reaches a steady-state abundance in ~2000 years track the abundances of stable isotopes through time?  This simply cannot be true.

I also suggested that the authors help the readers out with a clear explanation of the basics of 40Ar-39Ar dating. I provided an example of a paper on the same topic which provides the readers what they need to understand the paper.  It does not require a huge appendix.  It require 2-3 carefully written paragraphs. view

Author Response

Please, see attached file.
